# Olivetol’s Effects on Metabolic State and Gut Microbiota Functionality in Mouse Models of Alimentary Obesity, Diabetes Mellitus Type 1 and 2, and Hypercholesterolemia

**DOI:** 10.3390/biomedicines13010183

**Published:** 2025-01-13

**Authors:** Anastasia A. Zabolotneva, Katerina E. Popruga, Valentin V. Makarov, Sergei M. Yudin, Andrei M. Gaponov, Sergei A. Roumiantsev, Aleksandr V. Shestopalov

**Affiliations:** 1Department of Biochemistry and Molecular Biology, Faculty of Medicine, FSAEI HE N. I. Pirogov Russian National Research Medical University of MOH of Russia, 1 Ostrovitianov Str., 117997 Moscow, Russia; rumyantsev.sergey@endocrincentr.ru (S.A.R.); al-shest@yandex.ru (A.V.S.); 2Center for Strategic Planning and Management of Medical and Biological Health Risks of FMBA of Russia; Pogodinskya Str., h.10, b.1, 119121 Moscow, Russia; kpopruga@cspfmba.ru (K.E.P.); makarov@cspfmba.ru (V.V.M.); yudin@cspfmba.ru (S.M.Y.); 3V. A. Negovsky Research Institute of General Reanimatology, Federal Research and Clinical Center of Intensive Care Medicine and Rehabilitology, 141534 Moscow, Russia; zorba@yandex.ru

**Keywords:** alkylresorcinols, olivetol, obesity, metabolic disorders, mouse gut microbiota, *db*/*db* mice, *ldlr*(−/−) mice, NOD mice

## Abstract

Background: Disorders of glucose and lipid metabolism, such as obesity, diabetes mellitus, or hypercholesterolemia, can cause serious complications, reduce quality of life, and lead to increased premature mortality. Olivetol, a natural compound, could be proposed as a promising therapeutic agent for preventing, treating, or alleviating metabolic complications of such pathological conditions. Methods: In this study, the researchers conducted a broad parallel investigation of olivetol’s effects on metabolic state and gut microbiota functionality in mouse models of alimentary obesity, diabetes mellitus type 1 and 2, and hypercholesterolemia. Results: According to the results of the study, olivetol caused a lowering of body weight in C57Bl6 mice fed a high-fat diet and in *ldlr*(−/−) mice, decreased serum glucose levels in *db*/*db* mice, improved lipid metabolism in *ldlr*(−/−) mice, and prevented inflammatory infiltration of the pancreas and loss of insulin secretion in NOD mice. In addition, olivetol affected the composition and functional activity of gut microbiota communities, inducing an expansion of probiotic species such as *Akkermansia muciniphila* and *Bacteroides acidifaciens* and depleting the representation of pathobionts such as *Prevotella*, although olivetol supplementation did not influence the diversity or richness of the communities. Conclusions: These results suggest that olivetol is a promising therapeutic agent for preventing, treating, or alleviating the metabolic complications of obesity, diabetes mellitus type 1 and 2, and hypercholesterolemia; however, more investigations are required in order to attain a full understanding of its physiological effects.

## 1. Introduction

Metabolic disorders implicating abnormalities in glucose and lipid metabolism are a public health problem worldwide [1]. Obesity is a global health and socioeconomic problem. Over the past two decades, the number of obese or overweight people, among both adults and children, has more than doubled worldwide. Currently, about 30% of the world’s population is overweight [2]. Obesity is known to be associated with low-grade chronic systemic inflammation and may lead to insulin resistance, metabolic syndrome, or full-blown type 2 diabetes mellitus (T2D) [3,4].

Diabetes is anticipated to affect 463 million people worldwide (9.3% of adults aged 20 to 79), and by 2045, that number is expected to increase to 700 million [5]. T2D accounts for more than 90% of cases of diabetes mellitus [6].

Although type 1 diabetes mellitus (T1D) differs in etiology, it has many similar features to T2D in terms of the development of biochemical complications of pathological hyperglycemia. Glucose intolerance and accompanying chronic hyperglycemia have many negative effects on human health, including disturbances in lipid metabolism, early atherosclerosis, and increased risk of cardiovascular diseases, which are the main causes of human mortality [7]. Therefore, the desire to find the most effective approaches to the prevention and treatment of such pathological conditions is reasonable.

In recent years, we have observed an increasing interest in studying alkylresorcinol biological properties. Alkylresorcinols (ARs) are natural lipophilic compounds synthesized primarily by plants and bacteria [8]. Many positive metabolic effects of ARs, including anti-inflammatory [9], anti-carcinogenic [10], and anti-obesity effects [11], have been demonstrated in different studies [12].

For example, wheat bran-derived ARs have been found to improve glucose tolerance and insulin sensitivity by suppressing hepatic lipid accumulation and intestinal cholesterol absorption in mice, preventing the development of diet-induced obesity [13]. In a mouse model of T2D, wheat ARs demonstrated the ability to reduce blood glucose by improving glucose tolerance, altering the serum lipid profile, increasing liver glycogen content and glucokinase activity, enhancing the relative levels of GLUT2 mRNA in liver tissue and GLUT4 in epididymal adipose tissue, and simultaneously protecting pancreatic β-cells [14]. In a study of Liu et al. [15], it was established that AR intervention restored the serum GLP-1 level in mice compared to a group fed a high-fat diet (HFD) without AR supplementation. Furthermore, AR treatment alleviated HFD-induced damage to the ileal epithelium. In addition, AR treatment notably improved the abundance of SCFA-producing bacteria, such as *Bacteroides*, *Bifidobacterium*, and *Akkermansia*, in the mouse gut [15]. Increased levels of intestinal SCFAs improved the expression of short-chain fatty acid receptors (FFAR3) and the glucagon-like peptide-1 receptor (GLP-1R), which, in turn, improved the secretion of GLP-1 [15]. All these data indicate the high clinical potential of AR as a therapeutic agent for improving glucose homeostasis and treating obesity or its metabolic complications.

Olivetol (1,3-dihydroxy-5-pentylbenzene) is a natural alkylresorcinol that can be easily extracted from lichens [16]. Olivetol has been proposed for use as an anti-obesity agent for the treatment of disorders associated with hyperlipidemia [11]. In mouse and zebrafish models of high-fat diet-induced nutritional obesity, as well as in humans, olivetol has demonstrated the ability to reduce weight gain and improve lipid metabolism [11,13,17]. It has been established that high doses of olivetol reduce excessive fat accumulation due to the induction of lipid metabolism genes [11].

Dysmetabolic and dyslipidemic conditions are not only characterized by changes in systemic metabolism but are also associated with the development of a specific intestinal dysbiosis [18,19,20,21]. Increased intestinal permeability, microbial production of pro-inflammatory signaling molecules, and decreased diversity and richness of gut microbial communities are common signs of dysbiosis, which can lead to the deterioration of glucose and lipid metabolism [22].

In our study, we aimed to evaluate the effect of olivetol on the composition and functional activity of the intestinal microbiota in relation to the metabolic state in mouse models of nutritional obesity, T1D and T2D, and hypercholesterolemia, thereby assessing the therapeutic potential of olivetol in the treatment of such pathological conditions. To this end, we studied metabolic and hormonal characteristics, as well as gut microbiota composition and metabolic activity, in four mouse models fed different diets: C57Bl6spf mice fed a standard (SD) or high-fat diet (HFD) with or without olivetol (C5) supplementation to study the effects of olivetol on nutritional obesity; *db*/*db* (*lepr*−/−) mice fed an SD with or without C5 supplementation to study the effects of olivetol on T2D; NOD mice given SD with or without C5 supplementation to study the effects of olivetol on T1DM; and *ldlr*−/− mice fed an SD with or without C5 supplementation to study the effects of olivetol on hypercholesterolemia (Figure 1).

Because mice deficient in leptin receptors develop severe obesity independently of diet type, they can also represent a model of genetically determined obesity. However, in contrast to high-fat diet- or western diet-fed C57Bl6 mice, *db*/*db* mice more closely resemble human T2D [23].

## 2. Materials and Methods

### 2.1. Groups of Animals and Feeding

The linear C57BL/6spf mice (males, raised at the Pushchino Laboratory Animal Nursery, Pushchino, Russia) and knocked out (KO) mice (males, raised at the JAX-East and JAX-West Laboratory Animal Nurseries, Sacramento, CA, USA) used for this study were divided into ten groups, each containing an equal number of animals (Table 1).

For 1 week prior to formal study, mice were acclimated to housing conditions (22 °C, 55% humidity; 12 h/12 h light–dark cycle) in the SPF level animal center of I.M. Sechenov First Moscow State Medical University (Moscow, Russia). The animals had ad libitum access to sterile food and water. Following the adaptation period, the mice were divided into groups according to genotype and diet type (Table 1). At the beginning of the experiment. the mice were 8 weeks old and weighed 19 ± 2 g on average. To model alimentary obesity, C57BL/6SPF mice received either a high-fat diet (HFD) enriched with triglycerides, providing up to 30% of total calories (Altromin C 1090-30, Lage, Germany), or a standard diet (SD) (Altromin 1324 FORTI, Lage, Germany) starting at 8 weeks of age and lasting for 90 days until the end of the experiment. KO mice received a standard chow diet (Altromin 1324 FORTI, Lage, Germany) for the duration of the experiment.

Olivetol (Hangzhou ROYAL Import & Export Co., Ltd., Hangzhou, Zhejiang province, China, purity > 99%) was dissolved in a 1% water solution of pharmaceutical starch. Using an atraumatic intragastric tube, mice were given 0.2 mL of olivetol solution once a day for 90 days based on 2 mg/day dosage. The animals in the control groups were given a placebo (0.2 mL of a 1% water solution of pharmaceutical starch) in the same way.

At the end of the experiment, mice were anesthetized with isofurane (RWD Life Science, Chenzhen, Guangdong, China) and euthanized. Tissue samples were obtained in sterile conditions and immediately frozen in liquid nitrogen or taken for histology analysis. Colon samples were divided into 1 cm fragments, deposited into different sterile Eppendorf tubes, and stored at −80 °C until analysis. The colon contents samples were sent for high-throughput sequencing analysis (10 samples for each group).

All animal experiments were approved by the Ethics Committee for Animal Research, I.M. Sechenov First Moscow State Medical University, Russia (protocol number 96 from 2 September 2021).

### 2.2. Body Weight (BW), Food and Water Intake, Fasting Blood Glucose (FBG), Serum Triglyceride (TG), and Cholesterol Levels

Weekly observations of the mice’s BW, food, and water intake were carried out. Serum FBG, TG, and cholesterol levels were measured to evaluate the effect of olivetol on systemic metabolism. FBG was measured every 7 days for each group of mice; blood samples from tail nick sites were used for FBG analysis (GLUC3 test kit, cat. no. 04404483190, Roche, Switzerland). Before the FBG test, animals were fasted for 5 h with free access to water. TAG and cholesterol levels were measured at the end of the experiment after 90 days of feeding. Blood samples were collected from anesthetized mice via eyeball enucleation, kept at room temperature for 2 h, and centrifuged (1500 rpm at 4 °C) for 30 min to separate serum. TAG and cholesterol concentrations were determined using TRIGL and CHOL2 test kits, respectively (Roche, Switzerland; catalog numbers: 05171407188 and 03039773190, respectively), and a Cobas c311 blood analyzer (Roche, Switzerland) was also used.

### 2.3. Measurement of Serum Insulin, Leptin, Adiponectin, and Myostatin Levels

After an 8 h fasting period blood, samples were collected from anesthetized mice using the enucleation method. The serum was spun down at 8000× *g* for 8 min at 4 °C. The concentrations of insulin, leptin, adiponectin, and myostatin were determined using the ELISA method with the Mouse Insulin ELISA Kit (Abcam, Cambridge, UK; ab277390), Duo Set ELISA Development system Mouse Leptin (R&D System, Minneapolis, MN, USA; DY495-05), Adiponectin/Acrp30 DuoSet ELISA (R&D System; DY1119), and GDF-8/Myostatin DuoSet ELISA (R&DSystem; DY788-05), respectively, according to the manufacturer’s instructions.

### 2.4. High-Throughput Sequencing Analysis and Reconstruction of Intestinal Microbiota Metabolic Activity

Microbiota analysis was performed in the scientific research laboratory “Multiomics technologies of living systems” (Kazan, Russia), as was described previously [24,25]. Briefly, genomic DNA was extracted from mouse stool samples with the FastDNATM Spin Kit for Faeces (MP Biomedicals, Santa Ana, CA, USA). The bacterial 16S rRNA gene’s V3–V4 region was amplified using specific primers (see Appendix A). Samples were barcoded using the second round of PCR amplification with index primers. After PCR products purification, the samples were mixed in an identical mole ratio.

The libraries were then high-throughput sequenced (2 × 300 bp reads) (Illumina Miseq, Illumina, CA, USA). The raw reads were processed using QIIME2 v2023.7.0 [26] and PICRUSt2 v2.5.2 software [27] (accessed on 12 September 2023). Based on the sequencing data, the microbial metabolic pathways associated with the identified bacterial genomes were evaluated, and the most prevalent pathways were identified through multiple *t*-test analysis.

### 2.5. Histopathology of Mouse Liver, Pancreas, Skeletal Muscle, and Adipose Tissues

After fixation, the organs underwent dehydration, were immersed in paraffin, and were then sliced into sections 4–5 microns thick. These sections were stained with hematoxylin–eosin and analyzed under a light microscope at magnifications of ×10 and ×40. This study utilized histological equipment from Leica (Wetzlar, Germany). The organs were fixed for a minimum of 72 h following their collection. To improve fixation of organs, 24 h after the start of fixation, incisions were made in half of the mice (fixative: buffered formalin 10%, Leica). After fixation was completed, the organs were cut out for histological processing into histological cassettes. Wiring was carried out in isopropanol of increasing concentration (Isoprep, Biovitrum, St. Petersburg, Russia). The first three stations contained isopropanol in concentrations of 70%, 80%, and 90%, respectively. The next six stations contained absolute isopropanol. The infiltration time at the first station was 45 min; at subsequent stations, it was 1 h. After the wiring was completed, the organs were poured into paraffin blocks at an embedding station. Cooled paraffin blocks were cut using a microtome. The resulting sections were heated on a heating table for half an hour at a temperature of 40 °C, after which they were placed in a thermostat set to 60 °C overnight. The next day, the resulting sections were stained according to the following protocol: three stations with xylene (o-Xylene) for 3 min; three stations with alcohol for 3 min (with alcohol concentrations of 96%, 96%, and 70%, respectively); rinsing in distilled water for 3 min; rinsing in hematoxylin (Hematoxylin Gill II, Leica) for 10 min; rinsing in distilled water for 3 min; rinsing in eosin (Eosin Y, Leica) for 2 min; rinsing in distilled water for 3 min; rinsing via three stations with 95% alcohol concentration for 0.5, 2, and 2 min, respectively; rinsing with xylene–phenol mixture for 1 min; and rinsing via two stations with xylene for 2 min. Stained sections were covered with coverslips in an apparatus for mounting preparations in Sub-X Mounting Medium, Leica.

### 2.6. Immunohistochemistry

At the end of the experiment, after 90 days of feeding, mice were euthanized, and the pancreas was promptly fixed in formalin before being embedded in paraffin. Slides were deparaffinized and rehydrated, and then sodium citrate was used to retrieve antigens in preparation for immunohistochemistry labeling. Following an overnight incubation period in anti-insulin primary antibodies (CST 1:100, Danvers, MA, USA), sections were blocked in 3% bovine serum albumin, incubated for 2 h in secondary antibodies (Goat anti-Rabbit IgG (H + L) Cross-Adsorbed Secondary Antibody, P-2771 MP 1:400, USA), and mounted using prolonged anti-fade media containing DAPI.

Histological preparations were studied under Olympus CX41 (Tokyo, Japan) and LeicaDM 1000 (Wetzlar, Germany) microscopes with microphotography using a LeicaICC50 HD digital camera (Wetzlar, Germany) at 3 magnification levels: ×20 and ×40 lenses (drawings for calculating the final magnification of microphotos are presented in the Photo Appendix). Scanned images of the preparations were obtained on a Nikon Super Coolscan 8000 ED digital scanner (scale line on scans = 1 mm). Morphometric data were obtained using image processing and the analysis software Leica Application Suite (LAS) Version 4.9.0. The results of the immunohistochemical reactions were assessed according to the distribution of staining using a 3-point system, counting the number of immunopositive cells in 10 randomly selected fields of view at a magnification of ×400 (%): “–”, absent; “+”, weak (5–25% of cells, 1 point); “++”, moderate (25–50% of cells, 2 points); “+++”, severe (≥51% of cells, 3 points). The indicator for the marker (insulin) was determined as the ratio of the quantitative density of immunopositive cells to the quantitative density of cells of the islet of Langerhans stained with hematoxylin.

### 2.7. Statistical Data Analysis

Statistical analysis of the data was performed using non-parametric methods with the GraphPad Prism 10 v10.0.2 (171) software package. The results are presented as means with standard deviations. All in vivo experimental data were evaluated using Welch’s one-way analysis of variance (ANOVA), Welch’s unpaired *t*-test, or multiple Mann–Whitney tests, applying the two-stage step-up procedure (Benjamini, Krieger, and Yekutieli) with a false discovery rate of Q = 5%. *p*-values below 0.05 were deemed statistically significant (* *p* < 0.05, ** *p* < 0.01, *** *p* < 0.001, **** *p* < 0.0001).

## 3. Results

### 3.1. Body Weight and Hormonal Status

We investigated groups of mice (males; n = 10 for each group) fed a standard (SD) or high-fat diet (HFD) for 90 days. The dynamics of weight gain in different experimental groups is shown in Figure 2a,c,e,g. The final weights of the mice were compared at the end of the experiment (Table 2; Figure 2b,d,f,h).

Mice from both obesity models (alimentary and that caused by leptin receptor deficiency) showed significantly higher body weight compared to the C57Bl6 mice fed an SD (Table 1): the mean body weights of HFD-fed C57Bl6 and SD-fed *db*/*db* mice were, respectively, 22% and 95% higher than mean body weight of SD-fed C57Bl6 mice. These and several other metabolic, hormonal, and histological signs described below indicate the development of an obese state linked with dysmetabolic characteristics in these groups of mice. Olivetol supplementation (+C5) has led to lower weight gain in HFD-fed C57Bl6 (Figure 2b) and *ldlr* (−/−) mice (Figure 2h) compared to those who did not receive olivetol.

### 3.2. Fasting Blood Glucose (FBG), Serum Triacylglyceride (TAG) and Cholesterol Levels

FBG was measured every 7 days for every group of mice (Figure 3a–d). TAG and cholesterol levels were measured at the end of experiment, after 90 days of feeding. FBG levels were comparatively higher in *db*/*db* mice than in *db*/*db* mice administered olivetol (*p* < 0.05) (Figure 3b), whereas there were no differences in glucose levels in other groups (Figure 3a,c,d). However, the serum levels of TAG and cholesterol were significantly lower in *ldlr*(−/−) mice receiving olivetol (*p* < 0.01) (Figure 3d,e) than in *ldlr*(−/−) mice fed a standard diet without olivetol. Furthermore, olivetol supplementation led to decrease in TAG levels in C57Bl6 mice fed a standard diet but not those fed a high-fat diet (*p* < 0.0001) (Figure 3d). In contrast, olivetol supplementation caused an increase in TAG levels in NOD mice compared to NOD mice fed a standard diet (*p* < 0.05) (Figure 3d). In other groups, there were no differences in serum TAG and cholesterol levels.

### 3.3. Hormonal Status

To investigate the hormonal status of the mice, we compared the levels of insulin, leptin, adiponectin, myostatin, and TGF in the blood serum of the mice in all experimental groups after 90 days of feeding with or without olivetol supplementation (Figure 4a–d). We established that serum insulin and leptin levels were significantly higher in *db*/*db* mice compared to other mice (Figure 4a,b), which confirmed the genetically determined character of obesity, along with high glucose levels (Figure 3b), and confirmed the development of insulin resistance and glucose intolerance in these groups of mice. However, insulin levels were not influenced by olivetol supplementation (Figure 4a) in any of the groups, whereas leptin levels were lower in *db*/*db* mice and in HFD-fed C57Bl6 mice receiving olivetol (*p* < 0.05 and *p* < 0.0001, respectively) (Figure 4b) compared to their respective groups of mice that did not receive olivetol. Adiponectin levels were also lower in *db*/*db* mice and in HFD-fed C57Bl6 mice receiving olivetol (*p* < 0.0001) (Figure 4c) compared to their respective groups of mice that did not receive olivetol; whereas in NOD mice, olivetol supplementation caused an increase in adiponectin levels (Figure 4c). Myostatin levels were higher in the group of C57Bl6 mice fed an SD or HFD + olivetol compared to C57Bl6 mice fed an SD or HFD without olivetol (Figure 4d), and also in *db*/*db* mice receiving olivetol compared to *db*/*db* who did not.

### 3.4. Composition and Alpha Diversity of the Mouse Gut Microbiota

The diversity, richness, and composition of mouse colon microbiota were analyzed in all investigated groups of mice. Alpha diversity was used to express richness, diversity, and evenness (Figure 5a–c). Although we observed a decrease in the PD whole tree and Shannon indexes for microbiota in HFD-fed C57Bl6 mice compared to SD-fed mice, there were no differences in the alpha diversity indexes with respect to the dependence on olivetol supplementation (Figure 4a–c, Appendix A). The above results illustrate that olivetol intake in combination with an SD or HFD did not influence the diversity and richness of the mouse colon microbiota.

### 3.5. Taxonomy Composition of Gut Microbial Communities

To investigate the effects of olivetol supplementation on the mouse gut microbiota composition, we carried out metagenomic sequencing analysis of the colon microbiota of all the mice in ten different groups. At the phylum level, the microorganisms belonged primarily to five phyla *Firmicutes*, *Bacteroidetes*, *Proteobacteria*, *Verrucomicrobia*, and *Actinobacteria*. *Firmicutes* was the most dominant phylum in all the groups (Figure 6a–d). However, the *Firmicutes*/*Bacteroidetes* ratio did not change under different diet conditions.

At the phylum level, olivetol supplementation caused a significant increase in the representation of *Verrucomicrobia* in both HFD-fed C57Bl6 mice and in *ldlr*(−/−) mice (Figure 7a), *Proteobacteria* in HFD-fed C57Bl6 mice (Figure 7c), and *Bacteroidetes* in SD-fed C57Bl6 mice (Figure 7b). At the class level, we observed a significant increase in the representation of *Betaproteobacteria* and *Verrucomicrobiae* in HFD-fed C57Bl6 mice receiving olivetol (Figure 7e,h) compared to the HFD-fed mice who did not receive olivetol. In contrast, the representation of *Actinobacteria* and *Erysipelotrichi* was decreased in HFD-fed mice receiving olivetol (Figure 7d,g). Olivetol supplementation has also increased the representation of *Verrucomicrobiae* and *Erysipelotrichi* and decreased the representation of *Coriobacteriia* in *ldlr*(−/−) mice (Figure 7g,h). In NOD mice, we observed a decrease in the representation of *Betaproteobacteria* in the group that received olivetol (Figure 7e).

At the order level, we observed that olivetol increased the representation of *Erysipelotrichales* and *Verrucomicrobiales* and decreased the representation of *Coriobacteriales* in *ldlr* (−/−) mice (Figure 8a–c); it decreased the representation of *Erysipelotrichales* and *Bifidobacteriales* but increased the representation of *Verrucomicrobiales* in C57Bl6 mice fed an HFD (Figure 8a,c,e); and it decreased the representation of *Burkholderiales* in NOD mice (Figure 8d).

Analysis of mouse gut microbiota composition at the genus and species levels revealed that olivetol supplementation increased the representation of *Akkermansia* in C57Bl6 mice fed an HFD and in *ldlr*(−/−) mice (Figure 9a), decreased the representation of *Allobaculum* in C57Bl6 mice fed an HFD but increased it in *ldlr*(−/−) mice (Figure 9b), increased the representation of Parabacteroides in both *db*/*db* and *ldlr*(−/−) mice (Figure 9c), increased the representation of *Prevotella* in *db*/*db* mice but decreased it in NOD mice (Figure 9d), decreased the representation of *Bifidobacterium* in C57Bl6 mice fed an HFD (Figure 9e), decreased the representation of Bacteroides in *db*/*db* mice (Figure 9f), decreased the representation of *Adlercreutzia* in *ldlr*(−/−) mice (Figure 9g), decreased the representation of *Bacteroides acidifaciens* in *db*/*db* mice but increased it in *ldlr*(−/−) mice (Figure 9h), decreased the representation of *Bifidobacterium pseudolongum* in C57Bl6 mice fed an HFD (Figure 9i), and increased the representation of *Akkermansia muciniphila* in both C57Bl6 mice fed an HFD and in *ldlr*(−/−) mice (Figure 9j).

### 3.6. Reconstruction of the Metabolic Activity of the Mouse Gut Microbiota

To establish whether olivetol supplementation causes changes in intestinal microbiota metabolic activity, we performed a PICRUSt2 analysis based on metagenome sequencing data, which allowed us to estimate the representation of bacterial genes in the mouse gut microbiota communities.

Surprisingly, we have found that olivetol supplementation had almost no influence on predicted pathway abundance in the gut microbiota of *db*/*db*, NOD, and C57Bl6 (HFD-fed) mice: only a few pathways were differentially represented (Appendix A); whereas in *ldlr*(−/−) mice, 198 of 433 pathways were differentially represented (Appendix A); and in C57Bl6 (SD-fed) mice, 69 of 433 pathways were differentially represented (Appendix A); and 58 pathways were common for these groups. This fact may point to similar metabolic activity in the gut microbiota of C57Bl6 (SD-fed) and *ldlr*(−/−) mice in contrast to that of *db*/*db*, NOD, and C57Bl6 (HFD-fed) mice.

### 3.7. Histopathology of Mouse Liver, Pancreas, Skeletal Muscle, and Adipose Tissue

We conducted a morphological analysis of mouse tissues to evaluate the potential toxicity of olivetol. Changes found in these tissues serve as early indicators of harm to the animal’s health and are usually easier to detect than functional ones.

In the C57Bl6 mice fed a standard diet, the NOD, and the *ldlr* (−/−) mice, the histoarchitecture of the liver corresponded to the normal structure (Figure 10A(1),A(5),A(9), respectively). In C57Bl6 mice and *db*/*db* mice fed an HFD, the histological structure of the liver is disturbed due to fatty hepatosis (steatohepatosis) of a diffuse nature: about 2/3 of the hepatocytes, mainly in the perivenular and periportal zone of the hepatic lobule, were large, located at a considerable distance from each other, and some were vacuolated; their cytoplasm contained small-droplet and large-droplet lipid inclusions, indicating pronounced fatty degeneration of hepatocytes (Figure 10A(3),A(7), respectively).

In the groups that received olivetol, the liver retained a normal structure or displayed minor or mildly expressed sings of lipid infiltration (Figure 10A(2),A(4),A(6),A(8),A(10)). In C57Bl6 mice fed a high-fat diet and administered olivetol, we observed dusty lipid inclusions (Figure 10A(4)); in *db*/*db* mice, signs of steatohepatitis were moderately expressed and were predominantly focal in nature, closer to the portal tracts (Figure 10A(8)); in other groups, liver histoarchitecture corresponded normal structure. There were no signs of toxic or pro-inflammatory effects of olivetol on the liver in any group.

In C57Bl6 mice fed an SD and *db*/*db* mice, the pancreas structure corresponded to normal histoarchitecture: the pancreas had a lobular structure with exocrine and endocrine parts; the acini of the pancreas were located close to each other and passed into the excretory ducts; the islets of Langerhans were oval and elongated; and the insulocytes were large (Figure 10B(1),B(7), respectively).

In C57Bl6 mice fed a high-fat diet and *ldlr*(−/−) mice, the pancreas demonstrated signs of the fatty degeneration of acinocytes (steatopancreatitis) (Figure 10B(3),B(9)). In NOD mice, the islets of Langerhans were oval, elongated, and surrounded by inflammatory infiltration, predominantly lymphocytic (Figure 10B(5)). Furthermore, immunohistochemical analysis using antibodies to insulin revealed significant loss of insulin secretion (*p* < 0.001), which confirmed the development of T1DM in mice of this group (Figure 10B(5)): in preparations of the pancreas of mice from the control group, the proportion of insulin-positive cells was 62.8 ± 0.19%; whereas in NOD mice receiving olivetol, the number of insulin-positive cells was 76.1 ± 0.23%.

Administration of olivetol had a protective effect on the pancreas: in *db*/*db* mice, a slight increase in the number of islets and their structure was visually observed, and the islands were “more lush” (Figure 10B(8)); in NOD mice, we observed normal insulin secretion by beta cells (Figure 10B(6)); in the remaining groups, the pancreas had a normal structure without inflammatory and lipid infiltration (Figure 10B(2),B(4),B(10)).

In *db*/*db* mice, the layers of endomysium and perimysium in skeletal muscle tissue were expanded, and in some places, there was also a proliferation of white adipose tissue between muscle fibers, the size of which was visually reduced, indicating muscle dystrophy. At the same time, no inflammatory infiltration was detected (Figure 10C7).

Morphology analysis of skeletal muscle and adipose tissue in all resting groups, including those administered olivetol, has not revealed any pathological features or differences between groups with or without olivetol. The tissue structure corresponded to the normal histoarchitecture of skeletal muscle and adipose tissues in these groups (Figure 10C(1)–C(6),C(8)–C(10),D(1)–D(10)).

Thus, the administration of olivetol at a dose of 2 mg/day per mouse for 90 days did not cause any toxic effects, inflammation, or lipid infiltration of the tissues studied, as shown by morphological analysis. In addition, olivetol significantly improved the histological appearance of the liver and pancreas, preventing the development of steatosis and the loss of insulin secretion by beta cells.

## 4. Discussion

Olivetol is a short side chain representative of the homologous series of alkylresorcinols that are known to possess a wide range of biological effects, including anti-oxidant, hypolipidemic, and obesoprotective activity [12]. The mechanisms of olivetol’s action remain unclear; however, the anti-obesogenic and hypolipidemic properties of olivetol are associated with antagonistic activity on endocannabinoid receptors CB1, as well as with the inhibition of the activity of the HMG CoA reductase [11,28,29]. In addition, in silico molecular docking studies demonstrated the high binding efficiency of olivetol to cholecystokinin and GLP-1 receptors [11]. The versatility of olivetol effects is apparently not determined by its action only on certain types of receptors; it may also be associated with a modulating effect on the intestinal microbial community, which largely determines the characteristics of the host’s systemic metabolism [30,31].

We conducted a broad study of olivetol’s effects on the composition and functional activity of the intestinal microbiota, as well as on metabolic and hormonal status, in five mouse models, including T1D, T2D, hypercholesterolemia, and nutritional obesity, to evaluate its therapeutic potential in combating insulin resistance, dyslipidemia, and dysbiosis.

We did not find any data in the literature on the use of 5-pentylresorcinol in animal experiments before the start of this study. When selecting the dose, we relied on the results of our own preliminary testing of alkylresorcinols on animals, as well as on literature data on the experimental use of alkylresorcinols. For instance, 4-hexylresorcinol is known to cause similar positive metabolic effects and is proposed as a novel antidiabetic agent due to its hypoglycemic effects and stimulating activity on Glut4 expression, AMPK phosphorylation, and Histone H3 acetylation in the liver [32]. In studies involving an in vivo mouse model of *K. pneumoniae*-induced sepsis, the administration of 4-hexylresorcinol at a dose of 30–75 mg/kg for 5 days significantly increases the effectiveness of polymyxin treatment [32]. Therefore, the daily dose of 4-hexylresorcinol per mouse was approximately 1.0–3.5 mg. Enrichment of the diet with rye alkylresorcinols to 0.4–0.5% for 30 days inhibits the development of alimentary obesity, hypercholesterolemia, and fatty hepatosis in mice [13]. The approximate dose of alkylresorcinols in this experiment was 1.5–4 mg per day. Results from our pilot test of 5-pentylresorcinol showed that a single dose of 1.0 mg 5-pentylresorcinol in severely hyperglycemic NOD mice caused a slight decrease in blood glucose levels 5 h after administration, while a 10 mg dose caused a more than two-fold drop in glucose levels. Thus, the biological effects of alkylresorcinols are manifested in a wide range of concentrations from 0.01 to 10 mg/day. Considering the phenolic hydrophobic structure and the duration of use, we chose an average dosage of 2.0 mg per day. According to the results of our experiments, the administration of olivetol at a dose of 2 mg/day per mouse for 90 days did not cause any toxic effects. Furthermore, olivetol significantly improved the histological appearance of the liver and pancreas, preventing the development of liver fat infiltration and loss of insulin secretion by beta cells. Our data are in agreement with results of Reshma et al. [11]. In this study, olivetol’s effects were observed in both low-dose (0.001 g of olivetol dissolved in 1 mL distilled water) and high-dose (0.001 g of olivetol dissolved in 0.5 mL distilled water) olivetol exposition of zebrafish for 14 days for four hours each day. No toxic effects of olivetol were determined in a zebrafish alimentary obesity model. Furthermore, in this study, the researchers conducted in silico molecular docking analysis, which has shown that olivetol has strong bioavailability and exhibits effective pharmacokinetic characteristics such as gastrointestinal absorption and blood–brain barrier re-penetration. Thus, olivetol satisfies Lipinski’s rule and may be considered a drug-like molecule [11].

There are only limited data in the literature on olivetol or similar mechanisms underpinning alkylresorcinol’s action and toxicity. For instance, it was established in experiments with mice that hexylresorcinol doses of up to 125 mg/kg have no carcinogenic properties [33]. Moreover, these doses exhibited antimutagenic and antitumor properties [34]. However, in the study of Nikolaev et al. [35], it was demonstrated that doses of 4-hexylresorcinol 30 mg/kg/day were more effective for sepsis treatment than higher doses of 50 or 75 mg/kg/day. The authors suppose that side-effects due to overdosing of 4-HR had a negative impact on the recovery of infected animals. However, due to the absence of information on olivetol’s side effects and toxicity, preclinical trials are needed to evaluate its acute and chronic toxicity. This issue may be the subject of further research but is out of the scope of the present preliminary study of olivetol’s metabolic and microbiota modulatory effects.

Although olivetol did not change the alpha diversity of microbial communities, it caused significant changes in the representation of individual taxa in different groups of mice. Olivetol was also found to have different effects on the hormonal and metabolic status of mice in different groups, as shown in Table 3.

Obviously, olivetol decreased leptin levels in both C57Bl6 mice fed an HFD and *db*/*db* mice that developed obesity spontaneously. However, weight loss was observed in HFD-fed C57Bl6 mice only, not in *db*/*db* mice. This is an important observation, apparently explaining olivetol’s mechanisms of action.

Olivetol is known to antagonize CB1-receptors (CB1R), which are widely distributed both in the central nervous system and in the periphery [28,29]. On the other hand, in silico molecular studies have demonstrated the high binding efficiency of olivetol to GLP-1R [11]. Previously, it was demonstrated that GLP-1R and peripheral CB1R signaling interact bidirectionally to regulate energy homeostasis [36]. Inhibition of CB1R enhances the effects of GLP-1 on food consumption and fat mass, while GLP-1R signaling is partially necessary for the endocannabinoid’s influence on energy balance, as the weight loss resulting from peripheral CB1 blockade is diminished in GLP-1R knockout mice [36]. Thus, the ability of olivetol to block CB1R and to stimulate GLP-1R simultaneously makes it an effective hypophagic agent.

Studies in CB1-knockout mice (CB1-/-) have shown that the inactivation of CB1R results in hypophagia and leanness in both standard and high-fat diet-fed animals [37]. Thus, strong evidence of CB1R implication in eating behavior were provided. Subsequent studies involving peripherally restricted inverse CB1R agonists revealed that weight reduction in obese mice was achieved by reversing leptin resistance, decreasing food intake, and enhancing energy expenditure [38]. The authors proposed two main mechanisms of the inverse CB1R agonist to overcome leptin resistance: (i) it decreased the synthesis and release of leptin from white adipocytes by enhancing local sympathetic tone via presynaptic CB1R activation; and (ii) it boosted renal leptin clearance by increasing the expression and trafficking of the multiligand endocytic receptor megalin in proximal tubules, mediated by peroxisome proliferator-activated receptor-α (PPAR-α) and receptor-associated protein (RAP) chaperone.

Decreased leptin levels, in turn, enhance hypothalamic neurons’ sensitivity to anorexigenic signaling of leptin [39].

These findings may explain why the decrease in leptin levels in *db*/*db* mice under olivetol intervention has not led to weight loss. Olivetol, known as antagonist of CB1R, caused a decrease in leptin levels both in obese C57Bl6 and *db*/*db* mice; however, *db*/*db* mice were insensitive to leptin due to the lack of leptin receptors. Therefore, weight-lowering effects of olivetol are apparently mediated through the modulation of leptin-dependent hypothalamic neuron function, which defines eating behavior and energy expenditure.

The effects of olivetol on gut microbiota communities may also be influenced by its negative impact on CB1R. In a study by Mehrpouya-Bahrami et al., it was shown that blocking CB1R restored the abundance of *A. muciniphila* in a mouse model of diet-induced obesity, and these changes in microbiota composition occurred independently of calorie restriction and weight loss. Additionally, the study observed significant improvements in the expression of MUC2 and KLF4 genes in the colon of mice treated with a CB1R antagonist. The transcription factors KLF4 and MUC2 are involved in the differentiation of goblet cells, which is linked to mucin production in the colon. These findings suggest that *A. muciniphila* responds to increased mucin production in the host following the blockade of the CB1R, with mucin serving as a nutrient source for the growth of *A. muciniphila* [40].

These findings are in agreement with the observations of the present study. Olivetol administration caused a significant increase in the abundance of *A. muciniphila* in HFD-fed C57Bl6 and ldlr(−/−) mice. It is well known that obesity and dyslipidemia conditions are associated with alterations in gut microbiota composition and inflammatory background [41]. In the research conducted by Brandsma et al., it was shown that transferring the proinflammatory *casp1*(−/−) microbiota into *ldlr*(−/−) mice increases systemic inflammation and accelerates the process of atherogenesis [41]. At the same time, the administration of *A. muciniphila* has been established to protect against atherogenesis in *ApoE*^−/−^ mice [42,43]. Notably, the involvement of CB1R in atherogenesis was also described. The double knockout of LDLR and CB1R in mice has been shown to considerably attenuate vascular damage in hypercholesterolemic mice [43]. Thus, the ability of olivetol to stimulate *A. muciniphila* growth and promote SCFA production, apparently through the inhibition of CB1R, leads to an improvement in the metabolic state of the host.

*A. muciniphila* is known to play the role of a keystone species in the gut microbiota community [44]. Such species are essential for microbial communities, and their disruption can result in a significant alteration of the microbiome’s structure and function, leading to dysbiosis. In contrast, the restoration of keystone species abundance will lead to the improvement of microbiota functioning even without increasing its diversity. In our study, we observed significant changes in the metabolic activities of microbiota in HFD-fed C57BL6 and ldlr(−/−) mice under olivetol administration, whereas there were almost no alterations in other groups of mice. We also did not find any differences in the alpha diversity of the investigated microbial communities. The metabolic changes observed in HFD-fed C57BL6 and ldlr(−/−) mice were most likely associated with the increased abundance of A. muciniphila given the large number of common metabolic pathways altered in both groups. However, it is difficult to estimate the contribution of individual microbial pathways to systemic metabolism and gut microbiota functionality. This point may be addressed in further research.

### 4.1. Olivetol Effects in Mouse Models of Dietary-Induced Obesity and T2DM

Mice in two groups—C57Bl6 mice fed a high-fat diet and *db*/*db* mice—have developed obesity, as indicated by the comparison of body weight dynamics (Figure 2 and Table 1). Olivetol led to a decrease in leptin levels in both groups, as well as a decrease in adiponectin and an increase in myostatin levels. Leptin is one of the key regulators of metabolism and energy homeostasis, primarily released by adipose tissue and circulates at levels proportionate to acute changes in caloric intake or body fat mass percentage [45]. Obesity is strongly associated with leptin resistance, where leptin is unable to suppress hunger, increase energy expenditure, and control glucose and lipid metabolism [46,47]. Reducing leptin resistance is crucial for treating obesity; thus, the significant decrease in leptin levels under olivetol administration, along with the body weight loss observed in the HFD-fed group of mice, may indicate an improvement in leptin resistance. On the other hand, there were no differences in the body weights of *db*/*db* mice, which are genetically deficient in leptin receptors.

At the same time, the improved glucose metabolism observed in *db*/*db* mice may be associated with the protective effects of olivetol on the liver. In *db*/*db* mice receiving olivetol, as well as in HFD-fed mice receiving olivetol, hepatic steatosis was significantly less pronounced compared to the respective control group. Hepatic steatosis, which commonly accompanies obesity and metabolic syndrome [48], may progress to non-alcoholic fatty liver disease, fibrosis, or even cirrhosis [49]. Preventing fat infiltration of hepatocytes improves lipid and glucose metabolism and insulin resistance and lowers fasting blood glucose level [50].

The decrease in adiponectin levels and the simultaneous increase in myostatin levels in HFD-fed and *db*/*db* mice receiving olivetol may seem contradictory. However, this may indicate the inhibitory effects of olivetol on adipogenesis. Numerous in vitro and in vivo studies have demonstrated that myostatin suppresses preadipocyte differentiation, as evidenced by alterations in cellular shape and decreased expression of PPARγ and C/EBPα, which are recognized indicators of adipocyte differentiation and decreased lipid accumulation [51,52,53,54]. Additionally, earlier research has shown that adipogenic cells from *mstn*−/− mice exhibit increased protein expression of markers specific to mature adipocytes, such as adiponectin, AMP-activated protein kinase, PPARγ, and leptin [55,56,57]. Therefore, decreased adiponectin levels in HFD-fed and *db*/*db* mice receiving olivetol are likely associated with reduced adipogenesis, as adiponectin is a marker of terminally differentiated adipocytes [58].

Notably, olivetol treatment led to an expansion of *Akkermansia muciniphila* in the gut of HFD-fed C57Bl6 mice. *A. muciniphila* plays a crucial role in regulating glucose and energy metabolism. Its abundance is associated with improvements in insulin resistance, glucose tolerance, and metabolic endotoxemia [59,60] and is linked to lower adiposity and improved obesity outcomes [61,62].

### 4.2. Olivetol Effects in Mouse Models of Hyperlipidemia

Olivetol also caused a significant decrease in body weight in *ldlr*(−/−) mice (Figure 2b,h) and affected lipid metabolism by reducing serum levels of TAG and cholesterol (Figure 3d,e), which were—as expected—higher in the mouse model of hyperlipidemia compared to other groups [63]. Our data are consistent with previous research wherein olivetol treatment resulted in a reduction in total cholesterol levels, a simultaneous increase in high-density lipoprotein levels, a reduction in TAG levels, and weight loss in an adult zebrafish model of diet-induced obesity [11]. Olivetol has shown high effectiveness in decreasing the activity of the HMG CoA reductase and lowering liver transferases activity [11]. Thus, the hypolipidemic effects of olivetol in *ldlr*(−/−) mice may be mediated by inhibition of endogenous cholesterol synthesis and improved liver function.

Furthermore, olivetol treatment led to an increased abundance of *Allobaculum sp.*, *Bacteroides acidifaciens*, and *A. muciniphila* in gut microbiota of *ldlr*(−/−) mice. These species are short-chain fatty acid producers that have beneficial effects on the metabolic health of the host.

*B. acidifaciens* is known to prevent obesity and increase insulin sensitivity [64]. In previous studies, the expansion of *B. acidifaciens* was shown to lower of hepatic non-esterified fatty acid levels in high-fat diet-fed mice [65] and to protect against non-alcoholic steatohepatitis (NASH) development through acetate production [66]. Wang et al. showed that *B. acidifaciens* ameliorated liver injury by downregulating CD95 expression in hepatocytes and inhibiting hepatocyte apoptosis [67].

Numerous reports describe the protective properties of A. muciniphila in metabolic health and liver function. It has been shown that mice treated with A. muciniphila effectively cured fatty liver disease [68,69,70].

Therefore, the improvement in lipid metabolism observed in *ldlr*(−/−) mice under olivetol treatment may also be associated with expansion of “hepatoprotective” microbes in gut microbiota through modulation of “gut–liver” axis function.

### 4.3. Olivetol Effects in Mouse Models of T1DM

NOD mice are characterized by inflammatory infiltration of the perivascular duct and peri-islet regions of the pancreatic islets of Langerhans [71], accompanied by a progressive loss of insulin secretion. Histopathological analysis conducted in our study showed that NOD mice developed an inflammatory reaction in the islets of Langerhans and lost most of the insulin-secreting cells, while olivetol treatment prevented inflammatory damage to the pancreas, thus maintaining normal insulin secretion (Figure 10B5,B6). Although insulin levels did not differ among the studied groups, this is likely due to compensatory increases in insulin secretion by the beta cells of NOD mice. In NOD mice, which represent a model of T1D, we observed increased TAG and adiponectin levels. Additional studies are needed to investigate whether these effects are linked to the stimulation of adipogenesis or other metabolic changes, as well as whether such changes are protective in nature, preventing the development of diabetes.

It has been shown in the study of Clark et al. that HFD prevents autoimmune diabetes in NOD mice and maintains beta-cell mass and function by altering the gut microbiota, increasing T-regulatory cells, and reducing insulitis, although the mechanisms of such a defense remain unclear [72]. HFD is known to stimulate adipogenesis, as well as to increase TAG levels [73]. Therefore, olivetol may influence immune cell functioning through metabolic and hormonal signals, including those involved in lipid and adipose tissue metabolism. On the hand, we observed a significant decrease in the relative representation of *Prevotella* genus in the gut microbiota of NOD mice receiving olivetol compared to the microbiota of NOD mice that did not (Figure 9d). Increased representation of *Prevotella* is linked to enhanced mucosal inflammation mediated by T helper type 17 (Th17), consistent with *Prevotella*’s notable ability to induce Th17 immune responses in vitro. According to studies, *Prevotella* primarily activates Toll-like receptor 2, which causes antigen-presenting cells to produce Th17-polarizing cytokines, such as interleukin-23 (IL-23) and IL-1. *Prevotella* also stimulates the production of IL-8, IL-6, and CCL20 by epithelial cells, which can aid in the recruitment of neutrophils and mucosal Th17 immune responses. Mucosal inflammation mediated by *Prevotella* causes inflammatory mediators, bacteria, and bacterial products to spread throughout the body, potentially influencing the course of systemic diseases [74]. The decrease in *Prevotella* representation likely leads to the alleviation of the inflammatory response, thus preventing inflammatory damage to the pancreas.

## 5. Conclusions

For the first time, we conducted a broad parallel study of olivetol’s effects on metabolic state and gut microbiota functionality in mouse models of alimentary obesity, diabetes mellitus type 1 and 2, and hypercholesterolemia.

Although the effects of olivetol were multidirectional, it generally had a positive influence on systemic metabolism and the functioning of the liver and pancreas, as confirmed via histomorphology examination. Additionally, no toxic effects of olivetol on animal tissues were detected. According to the results of this study, olivetol caused a reduction in the body weight of HFD-fed C57Bl6 and *ldlr*(−/−) mice, decreased serum glucose levels in *db*/*db* mice, improved lipid metabolism in *ldlr*(−/−) mice, and prevented inflammatory infiltration of the pancreas and loss of insulin secretion in NOD mice. Furthermore, olivetol had microbiota-modulatory effects on gut microbiota communities, inducing the expansion of probiotic species, such as *Akkermansia muciniphila* and *Bacteroides acidifaciens*, while depleting the representation of pathobionts such as *Prevotella*, although olivetol supplementation did not influence the diversity or richness of the communities.

Our research has some limitations, such as the lack of studies on olivetol’s acute and chronic toxicity and its impact on inflammatory and immune status. These limitations were mainly associated with the limited amount of biomaterial for investigation and the broad scope of the study, which required the simultaneous analysis of different parameters in multiple groups of mice. However, these issues are expected to be investigated in further research.

Taking into account its high biological activity, its prebiotic properties, and the absence of pronounced toxic effects, olivetol can be considered a promising therapeutic agent for preventing, treating, or alleviating the metabolic complications of conditions such as alimentary obesity, diabetes mellitus type 1 and 2, and hypercholesterolemia. To our knowledge, there are no registered clinical trials of olivetol administration as a therapeutic agent for treating dysbiosis, obesity, and the related metabolic disorders or hyperlipidemia. However, considering previous studies and the present work, we believe it is advisable to conduct such trials.

## 6. Patents

A patent (RU2829635C1) was obtained as a result of the work reported in this manuscript.

## Figures and Tables

**Figure 1 biomedicines-13-00183-f001:**
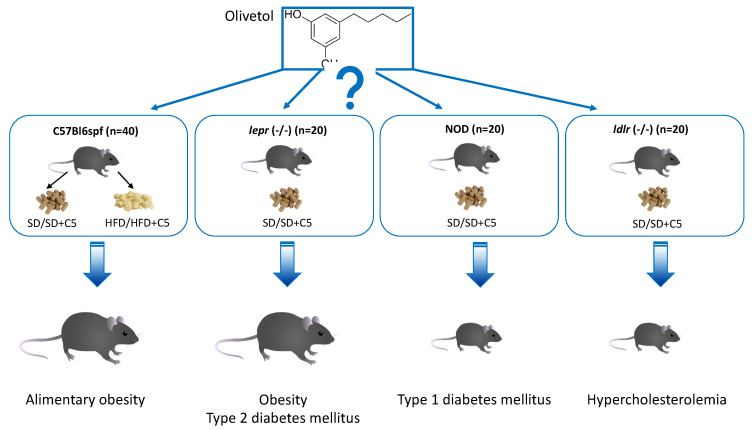
Experimental design. Four mouse models were selected to investigate the effects of olivetol on the metabolic state and gut microbiota composition of mice fed a standard diet (SD) or a high-fat diet (HFD). Each group, including control groups that did not receive olivetol, consisted of 10 mice. n indicates the total number of mice taken for the individual model studied in the experiment. HFD-fed C57Bl6 mice and *db*/*db* mice developed an obese phenotype compared with SD-fed C57Bl6 mice.

**Figure 2 biomedicines-13-00183-f002:**
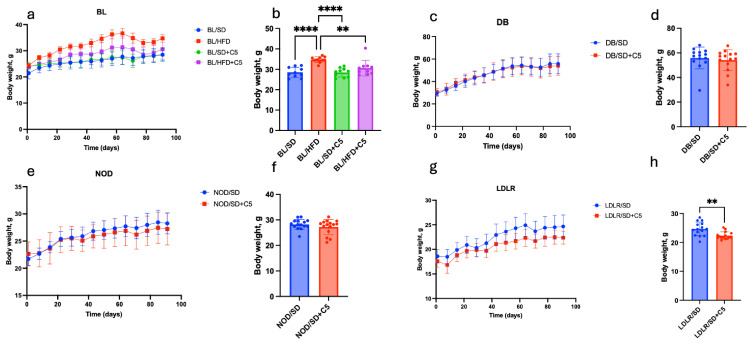
Mouse body weight dynamics during 90 days of the experiment: (**a**) C57Bl6 mice weights; (**b**) ANOVA followed by unpaired *t*-tests of C57Bl6 mice BW after 90 days of the experiment (** *p* < 0.01, **** *p* < 0.0001); (**c**) *db*/*db* mice weights; (**d**) unpaired *t*-test of *db*/*db* mice’s BW after 90 days of the experiment (no significant differences); (**e**) NOD mice weights; (**f**) unpaired *t*-test of NOD mice BW after 90 days of the experiment (no significant differences); (**g**) *ldlr*(−/−) mice weights; (**h**) unpaired *t*-test of *ldlr*(−/−) mice BW after 90 days of the experiment (** *p* < 0.01). BL/SD, C57Bl6 mice were fed a standard diet; BL/HFD, C57Bl6 mice were fed a high-fat diet; BL/SD + C5, C57Bl6 mice were fed a standard diet with olivetol supplementation; BL/HFD + C5, C57Bl6 mice were fed a high-fat diet with olivetol supplementation; LDLR/SD, *ldlr*(−/−) mice were fed a standard diet; LDLR/SD + C5, *ldlr*(−/−) mice were fed a standard diet with olivetol supplementation; NOD/SD, NOD mice were fed a standard diet; NOD/SD + C5, NOD mice were fed a standard diet with olivetol supplementation; DB/SD, *db*/*db* mice were fed a standard diet; and DB/SD + C5, *db*/*db* mice were fed a standard diet with olivetol supplementation.

**Figure 3 biomedicines-13-00183-f003:**
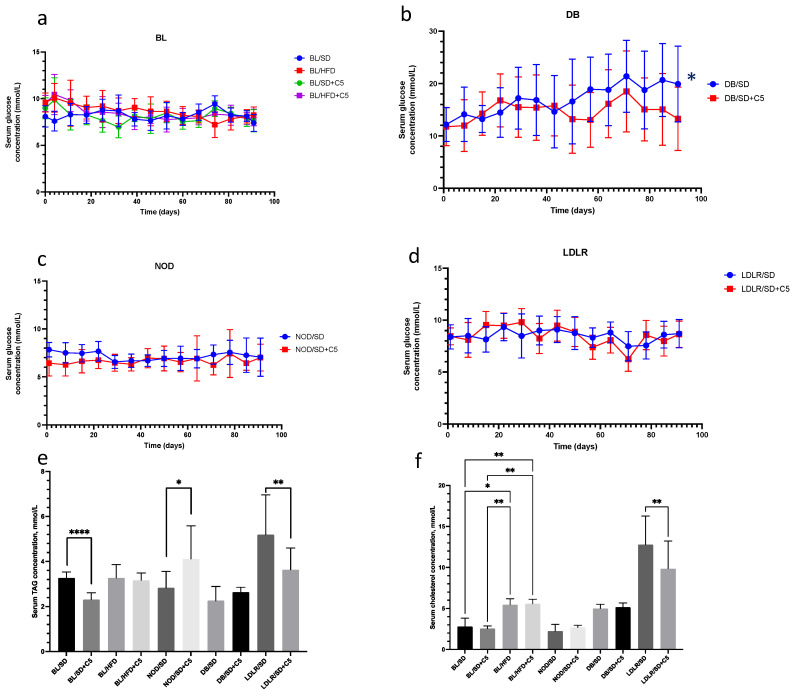
Results of the serum glucose dynamics during 90 days of experiment. (**a**–**d**), TAG (**e**), and cholesterol (**f**) levels were measured in the different experimental mice groups. Comparisons were carried out using unpaired *t*-test (* *p* < 0.05, ** *p* < 0.01, **** *p* < 0.0001). BL/SD, C57Bl6 mice were fed a standard diet; BL/HFD, C57Bl6 mice were fed a high-fat diet; BL/SD + C5, C57Bl6 mice were fed a standard diet with olivetol supplementation; BL/HFD + C5, C57Bl6 mice were fed a high-fat diet with olivetol supplementation; LDLR/SD, *ldlr*(−/−) mice were fed a standard diet; LDLR/SD + C5, *ldlr*(−/−) mice were fed a standard diet with olivetol supplementation; NOD/SD, NOD mice were fed a standard diet; NOD/SD + C5, NOD mice fed a standard diet with olivetol supplementation; DB/SD, *db*/*db* mice fed a standard diet; and DB/SD + C5, *db*/*db* mice were fed a standard diet with olivetol supplementation.

**Figure 4 biomedicines-13-00183-f004:**
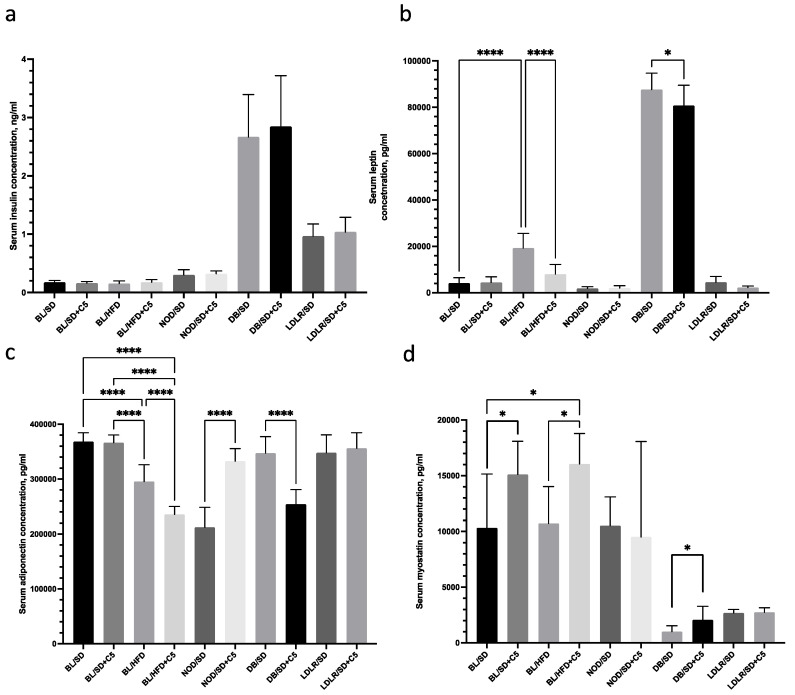
Results of the serum insulin (**a**), adiponectin (**b**), leptin (**c**), and myostatin (**d**) levels measured in the different experimental mice groups. Comparisons were carried out using unpaired *t*-tests (* *p* < 0.05, **** *p* < 0.0001). BL/SD, C57Bl6 mice were fed a standard diet; BL/HFD, C57Bl6 mice were fed a high-fat diet; BL/SD + C5, C57Bl6 mice were fed a standard diet with olivetol supplementation; BL/HFD + C5, C57Bl6 mice were fed a high-fat diet with olivetol supplementation; LDLR/SD, *ldlr*(−/−) mice were fed a standard diet; LDLR/SD + C5, *ldlr*(−/−) mice were fed a standard diet with olivetol supplementation; NOD/SD, NOD mice were fed a standard diet; NOD/SD + C5, NOD mice were fed a standard diet with olivetol supplementation; DB/SD, *db*/*db* mice were fed a standard diet; and DB/SD + C5, *db*/*db* mice were fed a standard diet with olivetol supplementation.

**Figure 5 biomedicines-13-00183-f005:**
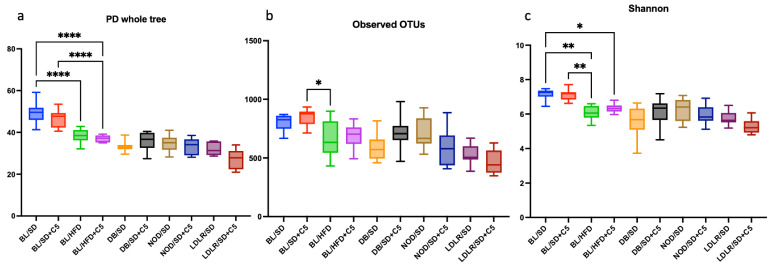
Results of unpaired *t*-tests for alpha diversity indexes ((**a**) for PD whole tree, (**b**) for Observed OTUs, and (**c**) for Shannon), indicating changes in the richness and diversity of the mouse colon microbiota. * *p* < 0.05, ** *p* < 0.01, **** *p* < 0.0001. BL/SD, C57Bl6 mice were fed a standard diet; BL/HFD, C57Bl6 mice were fed a high-fat diet; BL/SD + C5, C57Bl6 mice were fed a standard diet with olivetol supplementation; BL/HFD + C5, C57Bl6 mice were fed a high-fat diet with olivetol supplementation; LDLR/SD, *ldlr*(−/−) mice were fed a standard diet; LDLR/SD + C5, *ldlr*(−/−) mice were fed a standard diet with olivetol supplementation; NOD/SD, NOD mice were fed a standard diet; NOD/SD + C5, NOD mice were fed a standard diet with olivetol supplementation; DB/SD, *db*/*db* mice were fed a standard diet; and DB/SD + C5, *db*/*db* mice were fed a standard diet with olivetol supplementation.

**Figure 6 biomedicines-13-00183-f006:**
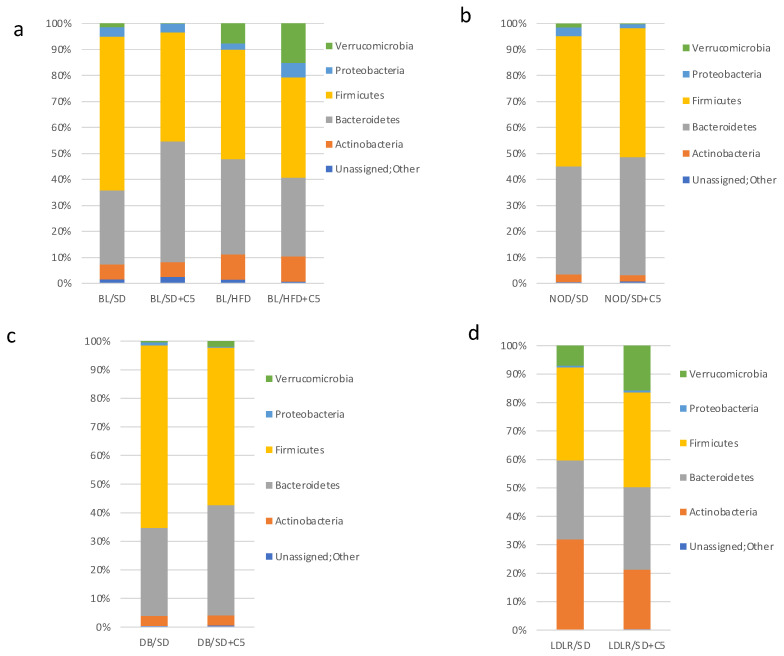
Mouse gut microbiota composition at phylum level in different groups of mice: (**a**) for C57Bl6 mice (BL/SD, C57Bl6 mice were fed a standard diet; BL/HFD, C57Bl6 mice were fed a high-fat diet; BL/SD + C5, C57Bl6 mice were fed a standard diet with olivetol supplementation; BL/HFD + C5, C57Bl6 mice were fed a high-fat diet with olivetol supplementation); (**b**) for NOD mice (NOD/SD, NOD mice were fed a standard diet; NOD/SD + C5, NOD mice were fed a standard diet with olivetol supplementation); (**c**) for *db*/*db* mice (DB/SD, *db*/*db* mice were fed a standard diet; and DB/SD + C5, *db*/*db* mice were fed a standard diet with olivetol supplementation); (**d**) for *ldlr*(−/−) mice (LDLR/SD, *ldlr*(−/−) mice were fed a standard diet; LDLR/SD + C5, *ldlr*(−/−) mice were fed a standard diet with olivetol supplementation).

**Figure 7 biomedicines-13-00183-f007:**
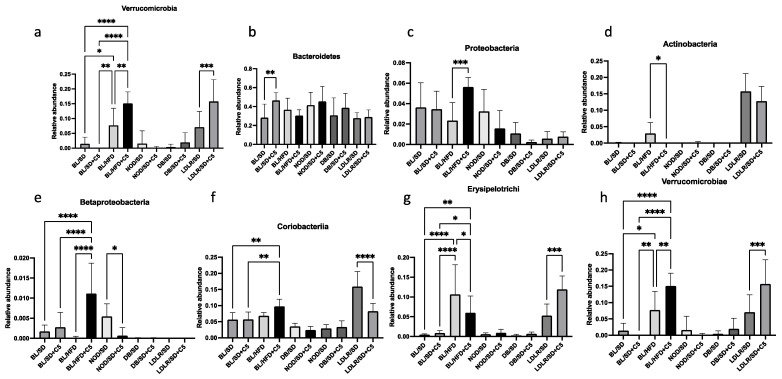
Effects of olivetol supplementation on the composition of the mouse gut microbiota community at phylum and class levels: (**a**) *Verrucomicrobia*; (**b**) Bacteroidetes; (**c**) *Proteobacteria*; (**d**) *Actinobacteria*; (**e**) *Betaproteobacteria*; (**f**) *Coriobacteriia*; (**g**) *Erysipelotrichi*; (**h**) *Verrucomicrobiae*. Comparisons were made via unpaired *t*-tests. * *p* < 0.05, ** *p* < 0.01, *** *p* < 0.001, **** *p* < 0.0001. BL/SD, C57Bl6 mice were fed a standard diet; BL/HFD, C57Bl6 mice were fed a high-fat diet; BL/SD + C5, C57Bl6 mice were fed a standard diet with olivetol supplementation; BL/HFD + C5, C57Bl6 mice were fed a high-fat diet with olivetol supplementation; LDLR/SD, *ldlr*(−/−) mice were fed a standard diet; LDLR/SD + C5, *ldlr*(−/−) mice were fed a standard diet with olivetol supplementation; NOD/SD, NOD mice were fed a standard diet; NOD/SD + C5, NOD mice were fed a standard diet with olivetol supplementation; DB/SD, *db*/*db* mice were fed a standard diet; and DB/SD + C5, *db*/*db* mice were fed a standard diet with olivetol supplementation.

**Figure 8 biomedicines-13-00183-f008:**
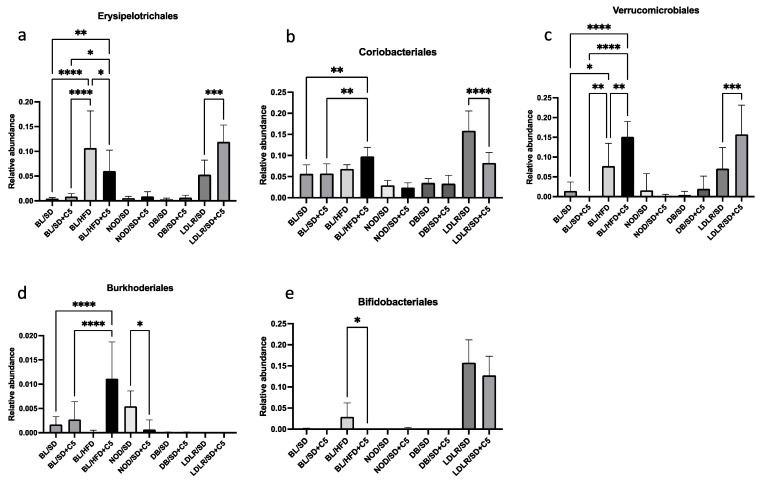
Effects of olivetol supplementation on the composition of the mouse gut microbiota community at the order level: (**a**) *Erysipelotrichales*; (**b**) *Coriobacteriales*; (**c**) *Verrucomicrobiales*; (**d**) *Burkhoderiales*; (**e**) *Bifidobacteriales*. Comparisons were made via unpaired *t*-tests. * *p* < 0.05, ** *p* < 0.01, *** *p* < 0.001, **** *p* < 0.0001. BL/SD, C57Bl6 mice were fed a standard diet; BL/HFD, C57Bl6 mice were fed a high-fat diet; BL/SD + C5, C57Bl6 mice were fed a standard diet with olivetol supplementation; BL/HFD + C5, C57Bl6 mice were fed a high-fat diet with olivetol supplementation; LDLR/SD, *ldlr*(−/−) mice were fed a standard diet; LDLR/SD + C5, *ldlr*(−/−) mice were fed a standard diet with olivetol supplementation; NOD/SD, NOD mice were fed a standard diet; NOD/SD + C5, NOD mice were fed a standard diet with olivetol supplementation; DB/SD, *db*/*db* mice were fed a standard diet; and DB/SD + C5, *db*/*db* mice were fed a standard diet with olivetol supplementation.

**Figure 9 biomedicines-13-00183-f009:**
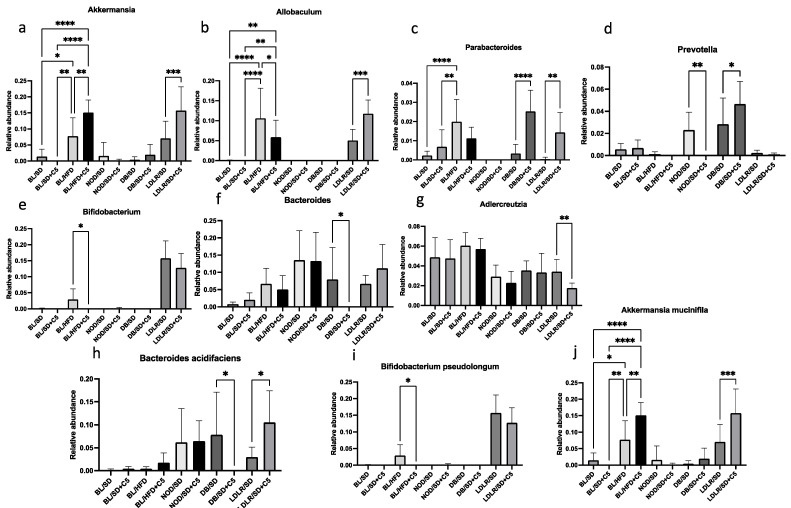
Effects of olivetol supplementation on the composition of the mouse gut microbiota community at genus and species level: (**a**) *Akkermansia*; (**b**) *Allobaculum*; (**c**) *Parabacteroides*; (**d**) *Prevotella*; (**e**) *Bifidobacterium*; (**f**) *Bacteroides*; (**g**) *Adlercreutzia*; (**h**) *Bacteroides acidifaciens*; (**i**) *Bifidobacterium pseudolongum*; (**j**) *Akkermansia muciniphila.* Comparisons were made via unpaired *t*-tests. * *p* < 0.05, ** *p* < 0.01, *** *p* < 0.001, **** *p* < 0.0001. BL/SD, C57Bl6 mice were fed a standard diet; BL/HFD, C57Bl6 mice were fed a high-fat diet; BL/SD + C5, C57Bl6 mice were fed a standard diet with olivetol supplementation; BL/HFD + C5, C57Bl6 mice were fed a high-fat diet with olivetol supplementation; LDLR/SD, *ldlr*(−/−) mice were fed a standard diet; LDLR/SD + C5, *ldlr*(−/−) mice were fed a standard diet with olivetol supplementation; NOD/SD, NOD mice were fed a standard diet; NOD/SD + C5, NOD mice were fed a standard diet with olivetol supplementation; DB/SD, *db*/*db* mice were fed a standard diet; and DB/SD + C5, *db*/*db* mice were fed a standard diet with olivetol supplementation.

**Figure 10 biomedicines-13-00183-f010:**
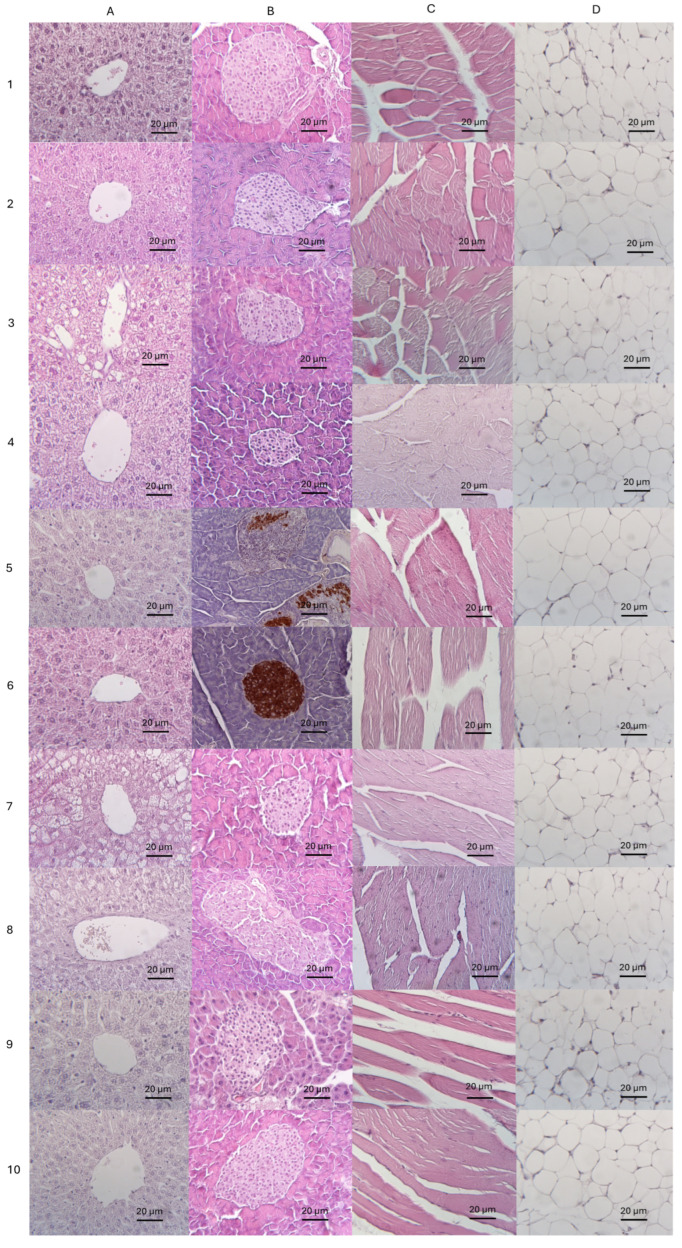
Tissue sections of mice receiving a diet without or with (+C5) olivetol supplementation (magnification ×200). Columns: (**A**) liver; (**B**) pancreas; (**C**) skeletal muscle; (**D**) adipose tissue. Rows: 1—BL/SD, C57Bl6 mice fed a standard diet; 2—BL/SD + C5, C57Bl6 mice fed a standard diet with olivetol supplementation; 3—BL/HFD, C57Bl6 mice fed a high-fat diet; 4—BL/HFD + C5, C57Bl6 mice fed a high-fat diet with olivetol supplementation; 5—NOD/SD, NOD mice fed a standard diet (beta cells were stained by antibodies to insulin); 6—NOD/SD + C5, NOD mice fed a standard diet with olivetol supplementation (beta cells were stained by antibodies to insulin); 7—DB/SD, *db*/*db* mice fed a standard diet; 8—DB/SD + C5, *db*/*db* mice fed a standard diet with olivetol supplementation; 9—LDLR/SD, *ldlr*(−/−) mice fed a standard diet; 10—LDLR/SD + C5, *ldlr*(−/−) mice fed a standard diet with olivetol supplementation. Tissue sections for all mice in groups can be provided by the corresponding author under request.

**Table 1 biomedicines-13-00183-t001:** Characteristics of the studied animals. SD, standard diet; HFD, high fat diet.

Type of Mice	Number of Animals in the Group	Diet Type
C57BL/6SPF	10	SD
C57BL/6SPF	10	HFD
C57BL/6SPF	10	SD + Olivetol
C57BL/6SPF	10	HFD + Olivetol
*ldlr*(−/−)	10	SD
*ldlr*(−/−)	10	SD + Olivetol
NOD	10	SD
NOD	10	SD + Olivetol
*db*/*db*	10	SD
*db*/*db*	10	SD + Olivetol

**Table 2 biomedicines-13-00183-t002:** Mean body weights of mice in the studied groups after 90 days of feeding of a specific diet. BL/SD, C57Bl6 mice were fed a standard diet; BL/HFD, C57Bl6 mice were fed a high-fat diet; BL/SD + C5, C57Bl6 mice were fed a standard diet with olivetol supplementation; BL/HFD + C5, C57Bl6 mice were fed a high-fat diet with olivetol supplementation; LDLR/SD, *ldlr*(−/−) mice were fed a standard diet; LDLR/SD + C5, *ldlr*(−/−) mice were fed a standard diet with olivetol supplementation; NOD/SD, NOD mice were fed a standard diet; NOD/SD + C5, NOD mice were fed a standard diet with olivetol supplementation; DB/SD, *db*/*db* mice were fed a standard diet; and DB/SD + C5, *db*/*db* mice were fed a standard diet with olivetol supplementation.

Group	Mean Body Weight After 90 Days of Feeding, g	Standard Deviation
BL/SD	28.5	2.50
BL/SD + C5	28.4	2.19
BL/HFD	34.8	1.37
BL/HFD + C5	30.6	3.82
NOD/SD	28.3	1.93
NOD/SD + C5	27.2	2.92
DB/SD	55.8	8.79
DB/SD + C5	54.1	8.48
LDLR/SD	24.7	2.35
LDLR/SD + C5	22.4	1.29

**Table 3 biomedicines-13-00183-t003:** Summary of olivetol’s effects on the metabolic and hormonal status of the investigated groups of mice. BW, body weight; GLC, serum glucose level; TAG, serum TAG level; CH, serum cholesterol level; INS, serum insulin level; LPT, serum leptin level; ADN, serum adiponectin level; MSTN, serum myostatin level. Arrows show increased or decreased levels of the investigated parameter in the corresponding group receiving olivetol supplementation compared to its respective control group that did not receive olivetol; absence of difference is indicated with a “-” sign. BL/SD, C57Bl6 mice were fed a standard diet; BL/HFD, C57Bl6 mice were fed a high-fat diet; 5—NOD/SD, NOD mice were fed a standard diet; DB/SD, *db*/*db* mice were fed a standard diet; and LDLR/SD, *ldlr*(−/−) mice were fed a standard diet.

	BW	GLC	TAG	CH	INS	LPT	ADN	MSTN
**BL/SD**	**-**	**-**	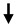	**-**	**-**	**-**	**-**	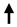
**BL/HFD**	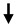	**-**	**-**	**-**	**-**	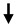	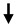	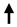
**NOD/SD**	**-**	**-**	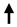	**-**	**-**	**-**	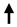	**-**
**DB/SD**	**-**	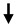	**-**	**-**	**-**	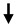	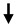	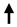
**LDLR/SD**	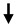	**-**	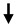	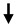	**-**	**-**	**-**	**-**

## Data Availability

The data presented in this study are available upon request from the corresponding author.

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
