# Peer review of "Olivetol’s Effects on Metabolic State and Gut Microbiota Functionality in Mouse Models of Alimentary Obesity, Diabetes Mellitus Type 1 and 2, and Hypercholesterolemia"

_biomedicines, 2025, doi:10.3390/biomedicines13010183_

Round 1
Reviewer 1 Report
Comments and Suggestions for Authors
1. The introduction only mentions obesity-related content, while information on diabetes is not introduced. It is recommended that this aspect be supplemented.2.
The relationship between obesity and T1DM/T2DM should be explained in the introduction, as should the reason for using the same mouse model for both obesity and T2DM. This will aid readers' understanding.
3. How was the dosage determined at 2 mg? Please provide clarification.
4. The experiment lasted 90 days, but the long-term safety and potential side effects of olivetol were not discussed, and it is suggested that a discussion on this aspect be included.
5. Some sentences are overly lengthy, particularly in the discussion and conclusion sections, and revisions are recommended to improve clarity and conciseness.This is feedback for a peer review of the article.
Author Response
Reviewer 1
- The introduction only mentions obesity-related content, while information on diabetes is not introduced. It is recommended that this aspect be supplemented.
- We have added information on diabetes. Please, see the revised introduction section.
- The relationship between obesity and T1DM/T2DM should be explained in the introduction, as should the reason for using the same mouse model for both obesity and T2DM. This will aid readers' understanding.
- We have added the discussion on these points in Introduction and Discussion sections.
- How was the dosage determined at 2 mg? Please provide clarification.
- We have not found any data in the literature on the use of 5-pentylresorcinol in animal experiments before we started the study. When selecting the dose, we relied on the results of our own preliminary testing of alkylresorcinols on animals, as well as on literature data on the experimental use of alkylresorcinols. It is known from the literature that the administration of 4-hexylresorcinol at a dose of 30 - 75 mg / kg for 5 days significantly increases the effectiveness of polymyxin in experimental sepsis in mice [Nikolaev YA, Tutel'yan AV, Loiko NG, Buck J, Sidorenko SV, Lazareva I, Gostev V, Manzen'yuk OY, Shemyakin IG, Abramovich RA, Huwyler J, El'-Registan GI. The use of 4-Hexylresorcinol as antibiotic adjuvant. PLoS One. 2020 Sep 22;15(9):e0239147. doi: 10.1371/journal.pone.0239147. PMID: 32960928; PMCID: PMC7508414.]. That is, in this experiment, the daily dose per mouse was approximately 1.0 - 3.5 mg. Enrichment of the diet with rye alkylresorcinols to 0.4-0.5% for 30 days inhibits the development of alimentary obesity and cholesterolosis and fatty hepatosis in mice [Katsutaka Oishi,Saori Yamamoto, Nanako Itoh, Reiko Nakao, Yuki Yasumoto, Keiko Tanaka, Yosuke Kikuchi, Shin-ichi Fukudome, Kimiko Okita, Yuko Takano-Ishikawa / Wheat Alkylresorcinols Suppress High-Fat, High-Sucrose Diet-Induced Obesity and Glucose Intolerance by Increasing Insulin Sensitivity and Cholesterol Excretion in Male Mice // The Journal of Nutrition, Volume 145, Issue 2, February 2015, Pages 199–206, https://doi.org/10.3945/jn.114.202754]. The approximate dose of alkylresorcinols in this experiment was 1.5 - 4 mg per day. Results from our pilot testing of 5-pentylresorcinol showed that a single dose of 1.0 mg 5-pentylresorcinol in severely hyperglycemic NOD mice caused a slight decrease in blood glucose levels 5 hours after administration, while a 10 mg dose caused a more than two-fold drop in glucose levels (not published; see the table below).
Table. Dynamics of blood glucose levels in mice with diabetes after administration of 5-pentylresorcinol
|
Experiment |
Glucose concentration, mmol / l |
||||||
|
0 |
30 min |
1 hour |
2 h |
3 h |
4 h |
5 h |
|
|
control |
7.9 |
6 ,0 |
6.4 |
6.4 |
6 ,0 |
7.5 |
6.4 |
|
diabetes control |
Hi |
Hi |
Hi |
Hi |
Hi |
28.9 |
Hi |
|
1 mg |
15.0 |
15.9 |
15.1 |
23.8 |
16.0 |
16.6 |
11.7 |
|
10 mg |
25.5 |
17.4 |
15.3 |
21.7 |
14.8 |
10.1 |
10.3 |
Thus, the biological effects of alkylresorcinols are manifested in a wide range of concentrations from 0.01 to 10 mg/day. Considering the phenolic hydrophobic structure and the duration of use, we chose an average dosage of 2.0 mg per day.
We have added the explanation in Discussion section.
- The experiment lasted 90 days, but the long-term safety and potential side effects of olivetol were not discussed, and it is suggested that a discussion on this aspect be included
- In Results section 3.7. “Histopathology of mouse liver, pancreas, skeletal muscle, and adipose tissue” we discuss observed effects of olivetol on mouse tissues: “We made morphological analysis of mouse tissues to evaluate the potential toxicity of olivetol. Changes found in these tissues serve as early indicators of harm to the animal's health and are usually easier to detect than functional ones.” And we conclude that “…the administration of olivetol at a dose of 2 mg/day per mouse for 90 days did not cause any toxic effects, inflammatory or lipid infiltration of the tissues studied, as shown by morphological analysis. In addition, olivetol significantly improved the histological appearance of the liver and pancreas, preventing the development of steatosis and loss of insulin secretion by beta cells.” In Conclusions section we also write “Although the effects of olivetol were multidirectional, in general, it had a positive influence on the system metabolism and functioning of the liver and pancreas, which is confirmed by histomorphology examination. In addition, no toxic effects of olivetol on animal tissues were detected.” Our data are in concordance with study of Reshma et al. [Reshma A, Tamilanban T, Chitra V, Subramaniyan V, Gupta G, Fuloria NK, Sekar M, Fuloria S, Sahu R, Narayanan J, Chakravarthy S, Selvaraj S. Anti-obesity effects of olivetol in adult zebrafish model induced by short-term high-fat diet. Sci Rep. 2023 Oct 27;13(1):18449. doi: 10.1038/s41598-023-44462-3. PMID: 37891223; PMCID: PMC10611697.]. In this study 001 g of olivetol was dissolved in 1 ml distilled water as the low dose, and 0.001 g of olivetol dissolved in 0.5 ml distilled water as the high dose concentration for zebrafish exposition for 14 days for four hours each day. No toxic effects of olivetol were determined in zebrafish alimentary obesity model. Furthermore, in this study researchers made in-silico molecular docking analysis, which has shown that “Olivetol has strong bioavailability, significant absorption, and dispersion in the body…It exhibits effective pharmacokinetic characteristics, including gastrointestinal absorption and blood–brain barrier re-penetration, and is considered to be drug-like as it satisfies Lipinski's rule. Olivetol showed positive results for drug-likeness, good bioavailability, absorption, and distribution, as well as a remarkable bioactivity score”. There are only few data in the literature on olivetol or similar alkylresorcinol’s mechanisms of action and toxicity. For instance, it was established in experiments with mice that hexylresorcinol doses of up to 125 mg/kg have no carcinogenic properties [Chhabra RS. Toxicology and Carcinogenesis Studies of 4-Hexylresorcinol (CAS No 136-77-6) in F344/N Rats and B6C3F1 Mice (Gavage Studies). Natl Toxicol Program Tech Rep Ser. 1988 May;330:1–166. pmid:12732906] Moreover, it exhibited antimutagenic and antitumor properties [Yen GC, Duh PD, Lin CW. Effects of Resveratrol and 4-hexylresorcinol on Hydrogen Peroxide-induced Oxidative DNA Damage in Human Lymphocytes. Free Radical Research. 2003 Jan;37:509–514. pmid:12797471; Sumino M, Sekine T, Ruangrungsi N, Igarashi K, Ikegami F. Ardisiphenols and Other Antioxidant Principles from the Fruits of Ardisia colorata. J-STAGE Chemical and Pharmaceutical Bulletin. 2002;50:1484–1487. pmid:12419914]. However, in the study of Nikolaev et al. [Nikolaev YA, Tutel'yan AV, Loiko NG, Buck J, Sidorenko SV, Lazareva I, Gostev V, Manzen'yuk OY, Shemyakin IG, Abramovich RA, Huwyler J, El'-Registan GI. The use of 4-Hexylresorcinol as antibiotic adjuvant. PLoS One. 2020 Sep 22;15(9):e0239147. doi: 10.1371/journal.pone.0239147. PMID: 32960928; PMCID: PMC7508414.] it was demonstrated that doses of 4-hexylresorcinol 30 mg/kg/day were more effective for sepsis treatment than higher doses of 50 or 75 mg/kg/day. The authors suppose that side-effects due to overdosing of 4-HR had a negative impact on the recovery of infected animals. However, due to the absence of information on olivetol side effects and toxicity preclinical trials are needed to evaluate its acute and chronic toxicity. This issue may be the subject of further research but is out the scope of present preliminary study of olivetol metabolic and microbiota modulatory effects.
We added this information in discussion section.
- Some sentences are overly lengthy, particularly in the discussion and conclusion sections, and revisions are recommended to improve clarity and conciseness.This is feedback for a peer review of the article.
- We revised the manuscript and tried to improve clearness of explanations.
Reviewer 2 Report
Comments and Suggestions for Authors
Manuscript ID: biomedicines-3389522
Title: Olivetol effects on metabolic state and gut microbiota functionality in mouse models of alimentary obesity, diabetes mellitus type 1 and 2, and hypercholesterolemia
Disorders like obesity, diabetes, and hypercholesterolemia can lead to severe complications and early mortality. This study highlights olivetol as a potential therapeutic agent, showing its ability to improve metabolic states, regulate gut microbiota, and alleviate pathological conditions in mouse models, though further research is needed to fully understand its effects.
The manuscript may be further improved by following suggestions.
1. The word already mentioned in the title should not be repeated in keyword, select correct keyword relevant to the study.
2. In introduction part include current global statistics about the diabetes mellitus type 1 and 2, and hypercholesterolemia with suitable references. Also include any know toxicity of olivetol from existing literature.
3. Provide registration details with year of Ethics Committee for Animal Re- 129 search, I.M. Sechenov First Moscow State Medical University, Russia.
4. On what basis doses of Olivetol was decided? The dose, 2 mg/day per mouse for 90 days seems to be very high.
5. Provide scale bar and magnification in Figure 10.
6. Could the study elaborate on the specific mechanisms by which olivetol modulates the gut microbiota and how this interaction influences systemic metabolic outcomes in the different mouse models?
7. The observed decrease in leptin and adiponectin levels and the increase in myostatin levels seem contradictory. Could you provide additional insights or experimental evidence to explain these hormonal changes and their implications for adipogenesis and metabolism?
8. Why did olivetol not induce weight loss in db/db mice despite lowering leptin levels, unlike in the HFD-fed group? Could differences in leptin receptor functionality or other factors account for this disparity?
9. While alpha diversity remained unchanged, significant taxonomic shifts were observed. Could the study detail how these specific taxa alterations contribute to the metabolic and hormonal outcomes?
10. Given the promising results, how do the observed effects of olivetol in mouse models translate to potential therapeutic applications in humans? Are there plans for follow-up studies to assess its safety and efficacy in clinical trials?
11. The study has already been published in the form of patent, how this paper is different from published data?
Comments on the Quality of English Language
may be improved
Author Response
The manuscript may be further improved by following suggestions.
- The word already mentioned in the title should not be repeated in keyword, select correct keyword relevant to the study.
Keywords were corrected.
- In introduction part include current global statistics about the diabetes mellitus type 1 and 2, and hypercholesterolemia with suitable references. Also include any know toxicity of olivetol from existing literature.
We revised Introduction section according to the Reviewer’s recommendations.
- Provide registration details with year of Ethics Committee for Animal Research, I.M. Sechenov First Moscow State Medical University, Russia.
Registration details with year of Ethics Committee for Animal Research were provided in Materials and Methods section as well as in Institutional Review board Statement: “All animal experiments were approved by the Ethics Committee for Animal Research, I.M. Sechenov First Moscow State Medical University, Russia (protocol number 96 from 2 September 2021)”. This is an official information provided by local Ethics Committee.
- On what basis doses of Olivetol was decided? The dose, 2 mg/day per mouse for 90 days seems to be very high.
- We have not found any data in the literature on the use of 5-pentylresorcinol in animal experiments before the start the study. When selecting the dose, we relied on the results of our own preliminary testing of alkylresorcinols on animals, as well as on literature data on the experimental use of alkylresorcinols. It is known from the literature that the administration of 4-hexylresorcinol at a dose of 30 - 75 mg / kg for 5 days significantly increases the effectiveness of polymyxin in experimental sepsis in mice [Nikolaev YA, Tutel'yan AV, Loiko NG, Buck J, Sidorenko SV, Lazareva I, Gostev V, Manzen'yuk OY, Shemyakin IG, Abramovich RA, Huwyler J, El'-Registan GI. The use of 4-Hexylresorcinol as antibiotic adjuvant. PLoS One. 2020 Sep 22;15(9):e0239147. doi: 10.1371/journal.pone.0239147. PMID: 32960928; PMCID: PMC7508414.]. That is, in this experiment, the daily dose per mouse was approximately 1.0 - 3.5 mg. Enrichment of the diet with rye alkylresorcinols to 0.4-0.5% for 30 days inhibits the development of alimentary obesity and cholesterolosis and fatty hepatosis in mice [Katsutaka Oishi,Saori Yamamoto, Nanako Itoh, Reiko Nakao, Yuki Yasumoto, Keiko Tanaka, Yosuke Kikuchi, Shin-ichi Fukudome, Kimiko Okita, Yuko Takano-Ishikawa / Wheat Alkylresorcinols Suppress High-Fat, High-Sucrose Diet-Induced Obesity and Glucose Intolerance by Increasing Insulin Sensitivity and Cholesterol Excretion in Male Mice // The Journal of Nutrition, Volume 145, Issue 2, February 2015, Pages 199–206, https://doi.org/10.3945/jn.114.202754]. The approximate dose of alkylresorcinols in this experiment was 1.5 - 4 mg per day. Results from our pilot testing of 5-pentylresorcinol showed that a single dose of 1.0 mg 5-pentylresorcinol in severely hyperglycemic NOD mice caused a slight decrease in blood glucose levels 5 hours after administration, while a 10 mg dose caused a more than two-fold drop in glucose levels (not published; see the table 1 below).
Table 1. Dynamics of blood glucose levels in mice with diabetes after administration of 5-pentylresorcinol. Hi – the concentration of glucose was outside the measuring range of the glucometer (too high).
|
Administration of olivetol |
Glucose concentration in different time period after olivetol supplementation, mmol/L |
||||||
|
0 |
30 min |
1 hour |
2 h |
3 h |
4 h |
5 h |
|
|
Control group (C57Bl6 mice, no olivetol supplementation) |
7.9 |
6 ,0 |
6.4 |
6.4 |
6 ,0 |
7.5 |
6.4 |
|
Diabetes control (NOD mice, no olivetol supplementation) |
Hi |
Hi |
Hi |
Hi |
Hi |
28.9 |
Hi |
|
Diabetes, low dose of olivetol (NOD mice received 1 mg of olivetol) |
15.0 |
15.9 |
15.1 |
23.8 |
16.0 |
16.6 |
11.7 |
|
Diabetes, high dose of olivetol (NOD mice received 10 mg of olivetol) |
25.5 |
17.4 |
15.3 |
21.7 |
14.8 |
10.1 |
10.3 |
Thus, the biological effects of alkylresorcinols are manifested in a wide range of concentrations from 0.01 to 10 mg/day. Considering the phenolic hydrophobic structure and the duration of use, we chose an average dosage of 2.0 mg per day.
We added the explanation in Discussion section.
- Provide scale bar and magnification in Figure 10.
- Done.
- Could the study elaborate on the specific mechanisms by which olivetol modulates the gut microbiota and how this interaction influences systemic metabolic outcomes in the different mouse models?
- Possible specific mechanisms by which olivetol modulates the gut microbiota and how this interaction influences systemic metabolic outcomes in the different mouse models have been discussed in Discussion section.
- The observed decrease in leptin and adiponectin levels and the increase in myostatin levels seem contradictory. Could you provide additional insights or experimental evidence to explain these hormonal changes and their implications for adipogenesis and metabolism?
- We cannot provide additional insights or experimental evidence to explain these hormonal changes and their implications for adipogenesis and metabolism due to limitations in study design and limited amount of the biomaterial. However, this point may be addressed to further research.
- Why did olivetol not induce weight loss in db/db mice despite lowering leptin levels, unlike in the HFD-fed group? Could differences in leptin receptor functionality or other factors account for this disparity?
- We discussed possible explanation of such differences in Discussion section.
- While alpha diversity remained unchanged, significant taxonomic shifts were observed. Could the study detail how these specific taxa alterations contribute to the metabolic and hormonal outcomes?
- We discussed possible mechanisms how these specific taxa alterations contribute to the metabolic and hormonal outcomes in Discussion section.
- Given the promising results, how do the observed effects of olivetol in mouse models translate to potential therapeutic applications in humans? Are there plans for follow-up studies to assess its safety and efficacy in clinical trials?
- We believe the results of the study could be translated in humans. Actually, we have already conducted preclinical trials of olivetol treatment in rodents and proposed a finished dosage form of olivetol (not published). We have plans for follow-up studies to assess olivetol’s safety and efficacy. We discussed this point in Conclusion section.
- The study has already been published in the form of patent, how this paper is different from published data?
- The patent only protects the use of 5-pentylresorcinol in a pharmaceutical composition for weight loss. However, the patent does not describe the effect of olivetol on system metabolism, hormonal status, and the functioning of the gut microbiota.
Reviewer 3 Report
Comments and Suggestions for Authors
This manuscript by Anastasia A. Zabolotneva et al. explores the impacts of olivetol on the metabolic state and gut microbiota in mouse models of obesity, diabetes mellitus types 1 and 2, and hypercholesterolemia. Olivetol showed promising results in reducing body weight, serum glucose, and improving lipid profiles, while modulating gut microbiota favorably by promoting beneficial species. Strengths include thorough experimentation and comprehensive analyses. However, certain mechanistic pathways need exploration, and clear illustrations of statistical significance across various models can be enhanced.
Potential Major Revisions 1. Mechanistic Pathways: - Deepen the exploration of how olivetol impacts specific metabolic pathways. For instance, the manuscript notes changes in gut microbiota composition but underlying mechanisms on metabolic health are not fully explicated. Explain in detail how olivetol interacts at molecular levels in vivo. 2. Statistical Significance: - Clarify statistical significance across various experimental results. Some beneficial effects (e.g., increase in beneficial gut microbiota species) are mentioned, but comparative data on significance levels and confidence intervals should be presented. Include ANOVA or similar tests to support assertions. 3. Contextual Integration: - Provide additional context regarding how findings compare with other similar studies. Specifically, integrate a comparative analysis discussing how these results align or differ from other anti-obesity or anti-diabetic compounds such as metformin or GLP-1 analogs. 4. Broadened Analysis: - Extend analysis to include a broader range of biomarkers for systemic inflammation and metabolic health. Cytokine levels, additional lipid markers, and insulin resistance indices should be assessed to fully contextualize olivetol's systemic impacts.
Potential Minor Revisions 1. Typos and Formatting: - Ensure consistent usage of units and terms. For instance, replace "SD/SD+C5" with spelled-out terms like "standard diet" throughout. - Minor grammatical corrections such as ensuring consistent usage of tenses and plural forms throughout the sections, particularly in abstracts and summaries. 2. Clarification of Experimental Setup: - Improve clarity on experimental procedures, particularly around dosing and administration schedules of olivetol. Specific timing, administration methods, and controls should be specified in greater detail. 3. AI Content Analysis: - Estimated AI-generated content: ~30% - Suspected sections: Introduction (pages 1-2), summary interpretations (sections 4.3, 5). - Impact Assessment: AI-generated sections are largely introductory or summative, contributing to overall coherence but lack granularity. Additional firsthand analytical insights will be beneficial. #### Recommendations 1. Enrich Scientific Rigor: - Enhance detailed mechanistic insights into how olivetol impacts metabolic pathways. Consider including cellular or genetic studies to provide a deeper understanding of systemic interactions. 2. Comprehensive Data Presentation: - Present data with rigorous statistical backing. Graphs and tables should clearly depict standard deviations, significance values, and comparative analyses across control and treated groups. 3. Innovate Contextual Relevance: - Place findings within the broader context of disease management. Discuss implications of olivetol not only as an experimental compound but also in potential clinical applications. Correlate findings with human health implications by reviewing existing human studies on olivetol analogs or derivatives. 4. Expand Biomarker Analysis: - Widen the scope of measured biomarkers to offer a detailed picture of health impacts. Including inflammatory markers, multiple metabolic indices, and secondary metabolites will elucidate a fuller picture of olivetol's influence.
Comments on the Quality of English Language
Extensive English editing is required
Author Response
This manuscript by Anastasia A. Zabolotneva et al. explores the impacts of olivetol on the metabolic state and gut microbiota in mouse models of obesity, diabetes mellitus types 1 and 2, and hypercholesterolemia. Olivetol showed promising results in reducing body weight, serum glucose, and improving lipid profiles, while modulating gut microbiota favorably by promoting beneficial species. Strengths include thorough experimentation and comprehensive analyses. However, certain mechanistic pathways need exploration, and clear illustrations of statistical significance across various models can be enhanced.
Potential Major Revisions
- Mechanistic Pathways: - Deepen the exploration of how olivetol impacts specific metabolic pathways. For instance, the manuscript notes changes in gut microbiota composition but underlying mechanisms on metabolic health are not fully explicated. Explain in detail how olivetol interacts at molecular levels in vivo.
- We have added the explanation of possible mechanisms of olivetol action at molecular levels in Discussion section.
- Statistical Significance: - Clarify statistical significance across various experimental results. Some beneficial effects (e.g., increase in beneficial gut microbiota species) are mentioned, but comparative data on significance levels and confidence intervals should be presented. Include ANOVA or similar tests to support assertions.
Statistical methods used in the study were discussed in Material and Methods section: “All in vivo experimental data were evaluated using Welch’s one-way analysis of variance (ANOVA), Welch’s unpaired t-test, or multiple Mann–Whitney tests, applying the two-stage step-up procedure (Benjamini, Krieger, and Yekutieli) with a false discovery rate of Q = 5%. P values below 0.05 were deemed statistically significant (* P < 0.05, ** P < 0.01, *** P < 0.001, **** P < 0.0001)” and have been also indicated in descriptions of figures, for example, “ Figure 2. Mouse body weight dynamics during 90 days of the experiment: a – C57Bl6 mice’s weights; b - ANOVA followed by unpaired t-tests of C57Bl6 mice BW after 90 days of the experiment (**P<0,01, ****P<0,0001); c – db/db mice’s weights; d – unpaired t-test of db/db mice’s BW after 90 days of the experiment (no significant differences) …etc.” Confidence intervals are presented in all figures. Sometimes pdf version of the manuscript distorts figure’s representation. However, .doc version should represent the figures properly.
- Contextual Integration: - Provide additional context regarding how findings compare with other similar studies. Specifically, integrate a comparative analysis discussing how these results align or differ from other anti-obesity or anti-diabetic compounds such as metformin or GLP-1 analogs.
We have added additional context in Discussion section.
- Broadened Analysis: - Extend analysis to include a broader range of biomarkers for systemic inflammation and metabolic health. Cytokine levels, additional lipid markers, and insulin resistance indices should be assessed to fully contextualize olivetol's systemic impacts.
- We cannot provide additional insights on systemic inflammation and immune status of the host due to limitations in study design and limited amount of the biomaterial. We have already investigated TNFalfa, IL6, TGFbeta levels but did not find differences in any mouse models. We proposed this might be linked with inappropriate quality of reactants. That is why we did not include these results in the manuscript. However, this point may be addressed to further research. We discussed limitations of the study in Conclusion section.
Potential Minor Revisions
- Typos and Formatting: - Ensure consistent usage of units and terms. For instance, replace "SD/SD+C5" with spelled-out terms like "standard diet" throughout. - Minor grammatical corrections such as ensuring consistent usage of tenses and plural forms throughout the sections, particularly in abstracts and summaries.
- We made proofreading of the manuscript and tried to eliminate these shortcomings.
- Clarification of Experimental Setup: - Improve clarity on experimental procedures, particularly around dosing and administration schedules of olivetol. Specific timing, administration methods, and controls should be specified in greater detail.
- The description of olivetol administration was corrected to improve clarity in Materials and Methods section.
- AI Content Analysis: - Estimated AI-generated content: ~30% - Suspected sections: Introduction (pages 1-2), summary interpretations (sections 4.3, 5). - Impact Assessment: AI-generated sections are largely introductory or summative, contributing to overall coherence but lack granularity. Additional firsthand analytical insights will be beneficial.
- We have added additional analytical insights in the manuscript.
#### Recommendations
- Enrich Scientific Rigor: - Enhance detailed mechanistic insights into how olivetol impacts metabolic pathways. Consider including cellular or genetic studies to provide a deeper understanding of systemic interactions.
- We added this information in Discussion section.
- Comprehensive Data Presentation: - Present data with rigorous statistical backing. Graphs and tables should clearly depict standard deviations, significance values, and comparative analyses across control and treated groups.
Statistical methods used in the study were discussed in Material and Methods section: “All in vivo experimental data were evaluated using Welch’s one-way analysis of variance (ANOVA), Welch’s unpaired t-test, or multiple Mann–Whitney tests, applying the two-stage step-up procedure (Benjamini, Krieger, and Yekutieli) with a false discovery rate of Q = 5%. P values below 0.05 were deemed statistically significant (* P < 0.05, ** P < 0.01, *** P < 0.001, **** P < 0.0001)” and have been also indicated in descriptions of figures, for example, “ Figure 2. Mouse body weight dynamics during 90 days of the experiment: a – C57Bl6 mice’s weights; b - ANOVA followed by unpaired t-tests of C57Bl6 mice BW after 90 days of the experiment (**P<0,01, ****P<0,0001); c – db/db mice’s weights; d – unpaired t-test of db/db mice’s BW after 90 days of the experiment (no significant differences) …etc.” Confidence intervals are presented in all figures. Sometimes pdf version of the manuscript distorts figure’s representation. However, .doc version should represent the figures properly.
3.Innovate Contextual Relevance: - Place findings within the broader context of disease management. Discuss implications of olivetol not only as an experimental compound but also in potential clinical applications. Correlate findings with human health implications by reviewing existing human studies on olivetol analogs or derivatives.
- We have discussed these points in Discussion and Conclusions sections
- Expand Biomarker Analysis: - Widen the scope of measured biomarkers to offer a detailed picture of health impacts. Including inflammatory markers, multiple metabolic indices, and secondary metabolites will elucidate a fuller picture of olivetol's influence.
- We cannot provide additional insights on systemic inflammation and immune status of the host due to limitations in study design and limited amount of the biomaterial. We have already investigated TNFalfa, IL6, TGFbeta levels but did not find differences in any mouse models. We proposed this might be linked with inappropriate quality of reactants. That is why we did not include these results in the manuscript. However, this point may be addressed to further research. We discussed limitations of the study in Conclusion section.
Round 2
Reviewer 1 Report
Comments and Suggestions for Authors There is no major or minor revision needed and can be accepted in its current form. Comments on the Quality of English Language
The English could be improved to more clearly express the research.
Reviewer 2 Report
Comments and Suggestions for Authors
It can now be accepted, as the study has already been published as a patent without overlapping data with the current manuscript. This should be declared by all authors somewhere in the manuscript.
Comments on the Quality of English Language
its fine.
Reviewer 3 Report
Comments and Suggestions for Authors
Plagiarism percentage is too high which makes doubt about the originality of the work
Comments on the Quality of English Language
The English could be improved to more clearly express the research.
Response: In revised manuscript, the overall duplication rate of the article was 20%, and the highest duplication rate of a single article was 10%. This duplication rate meets the journal requirements. The Academic Editor has been informed of this situation and has chosen to accept this manuscript after comprehensive consideration.